# A PIP$_2$ substitute mediates voltage sensor-pore coupling in KCNQ activation

Yongfeng Liu [1,4], Xianjin Xu[2,4], Junyuan Gao[3,4], Moawiah M. Naffaa[1], Hongwu Liang[1], Jingyi Shi[1], Hong Zhan Wang [3], Nien-Du Yang [1], Panpan Hou [1], Wenshan Zhao[1], Kelli McFarland White[1], Wenjuan Kong[1], Alex Dou[1], Amy Cui[1], Guohui Zhang[1], Ira S. Cohen[3✉], Xiaoqin Zou[2✉] & Jianmin Cui [1✉]

KCNQ family K$^+$ channels (KCNQ1-5) in the heart, nerve, epithelium and ear require phosphatidylinositol 4,5-bisphosphate (PIP$_2$) for voltage dependent activation. While membrane lipids are known to regulate voltage sensor domain (VSD) activation and pore opening in voltage dependent gating, PIP$_2$ was found to interact with KCNQ1 and mediate VSD-pore coupling. Here, we show that a compound CP1, identified in silico based on the structures of both KCNQ1 and PIP$_2$, can substitute for PIP$_2$ to mediate VSD-pore coupling. Both PIP$_2$ and CP1 interact with residues amongst a cluster of amino acids critical for VSD-pore coupling. CP1 alters KCNQ channel function due to different interactions with KCNQ compared with PIP$_2$. We also found that CP1 returned drug-induced action potential prolongation in ventricular myocytes to normal durations. These results reveal the structural basis of PIP$_2$ regulation of KCNQ channels and indicate a potential approach for the development of anti-arrhythmic therapy.

[1] Department of Biomedical Engineering, Center for the Investigation of Membrane Excitability Disorders, Cardiac Bioelectricity and Arrhythmia Center, Washington University in Saint Louis, Saint Louis, MO 63130, USA. [2] Dalton Cardiovascular Research Center, Department of Physics and Astronomy, Department of Biochemistry, Institute for Data Science & Informatics, University of Missouri, Columbia, MO 65211, USA. [3] Department of Physiology and Biophysics, and Institute for Molecular Cardiology, Stony Brook University, Stony Brook, NY 11794, USA. [4]These authors contributed equally: Yongfeng Liu, Xianjin Xu, Junyuan Gao. ✉email: ira.cohen@stonybrook.edu; ZouX@missouri.edu; jcui@wustl.edu

Voltage-gated KCNQ potassium channels (KCNQ1–5, also known as $K_V$7.1-5) are important in regulating cardiac action potential duration[1–3], modulating neuroexcitability[4–6], and maintaining endolymph $K^+$ homeostasis in the inner ear[7,8]. The KCNQ channels are activated by membrane depolarization, but they all share an important feature in that their activation also requires the membrane lipid, phosphatidylinositol 4,5-bisphosphate ($PIP_2$)[9–12]. KCNQ1 and its regulatory subunit KCNE1 form the $I_{Ks}$ channel in cardiac myocytes that terminates action potentials and regulates heart rhythm[13]. Congenital mutations in KCNQ1 that alter $I_{Ks}$ channel function are associated with cardiac arrhythmias[3,14,15]. Patients with mutations associated with $PIP_2$ regulation of KCNQ1 may exhibit a high risk for life-threatening events[16–18]. The heterotetramer $K^+$ channel formed by KCNQ2 and KCNQ3 was identified as carrying neuronal M currents[4]. Homotetramer KCNQ2 and KCNQ5 channels were suggested to contribute to M currents as well[19,20]. In neurons, muscarinic stimulation of Gq protein-signaling pathways hydrolyzes $PIP_2$. The reduction of membrane content of $PIP_2$ results in the decrease of M currents and the enhancement of neuronal excitability[6,12,21]. Congenital mutations that reduce M currents are associated with epilepsy and deafness[12].

$PIP_2$ has been shown to mediate voltage-dependent gating of KCNQ channels. In tetrameric KCNQ channels, each subunit contains 6 transmembrane segments S1–S6. Transmembrane segments S1–S4 form the voltage-sensor domain (VSD), and the pore is formed by S5–S6 from all four subunits. In the channel structure, four VSDs are located at the periphery of the central pore[22]. In voltage-gated $K^+$ channels, membrane depolarization activates the VSDs by moving the S4 segment to the extracellular side. The conformational changes in the VSD are then coupled to the pore for channel opening. In KCNQ1 channels, numerous experiments detected that the VSD activates in two steps, first to an intermediate state and then to the activated state[23–27]. In both states, VSD activation opens the pore, resulting in the intermediate open (IO) and activated open (AO) states[23,24,27,28]. The VSD activation opens the pore via two distinctive sets of interactions. The first interactions are between the S4–S5 linker (S4–S5L) and the cytosolic end of S6 within the same subunits, and the second interactions are between the neighboring subunits involving S4 and S4–S5L in one subunit and S5 and S6 in the other[27]. Ion channels interact with their lipid environments during channel function. Specific lipid-channel interactions in voltage-gated $K^+$ ($K_V$) channels have been reported to alter VSD activation or pore opening[29–32]. On the other hand, in KCNQ1 channels, $PIP_2$ was found to be required for VSD-pore coupling. When $PIP_2$ is depleted from the membrane, the pore cannot open, even though the VSD remains activated by membrane depolarization[18]. A site at the interface between the VSD and pore was identified for $PIP_2$ to interact and mediate VSD-pore coupling[9,33–36].

The mechanism by which $PIP_2$ mediates VSD-pore coupling in KCNQ1 remains to be understood. The structural determinants of the channel that are involved in this mechanism are not known. Further, the structural feature of the $PIP_2$ molecule that is important for its activity is also not clear. In this study, we identified a small molecule, CP1, by screening in silico using compounds resembling the $PIP_2$ head group to dock onto the structure model of KCNQ1, targeting the previously identified $PIP_2$-binding site[18]. We found that CP1 could substitute for $PIP_2$ to recover KCNQ1 currents that were abolished by depletion of membrane $PIP_2$. Similar to $PIP_2$, CP1 activated KCNQ1 channels by enhancing VSD-pore coupling. The compound also activated KCNQ2 and KCNQ3, but to an extent less than that seen for KCNQ1. On the other hand, CP1 showed significant differences from $PIP_2$ in its characteristics and mechanism of activating the

KCNQ channels. These results provide insights into $PIP_2$ and CP1 activation of KCNQ channels, and a basis for future development of antiarrhythmic drugs that target KCNQ1 channels.

## Results

KCNQ channels require $PIP_2$ for activation[9–11,18,37]. Our previous studies show that $PIP_2$ mediates the coupling between voltage-induced VSD movements and pore opening in KCNQ1, and it may associate with the channel protein at the interface between the VSD and the pore for this function[18]. To further understand the effects of $PIP_2$ on KCNQ channel activation and how the VSD-pore coupling is influenced by $PIP_2$, we investigated if other compounds resembling $PIP_2$ can mediate the interactions between the VSD and the pore for channel activation. We screened, in silico, a subset of the Available Chemical Database (ACD, Molecular Design Ltd.), in which each compound has a formal charge of 2 ($\sim 10^4$ compounds), targeting the $PIP_2$ site using both the structure-based and ligand-based methods. In the structure-based screening, compounds in the ACD subset were ranked by their binding scores calculated using an in-house docking software MDock[38–40]. In the ligand-based screening, compounds were ranked by their similarity scores with the head group of $PIP_2$, calculated using a 3D molecular similarity program ShaEP[41]. The details are described in the "Methods" section.

From the primary screen, a compound, CP1, was found in the top 2% for both structure-based and ligand-based strategies, which contains two sulfates that may mimic the head group of $PIP_2$ (Fig. 1a). Docking of $PIP_2$ and CP1 onto the homology model of human KCNQ1 shows that the two molecules interact with KCNQ1 in the same pocket formed by the S4–S5L and the C terminus of S6 (S6C) (Fig. 1b). The interaction between the S4–S5 linker and S6C is important for the VSD activation to be coupled to pore opening in KCNQ1[18,27] and other $K_V$ channels[42,43]. While CP1 and $PIP_2$ interact with a distinct set of KCNQ1 residues (see Supplementary Fig. 1), some of these interacting residues, including K354 and K358, are shared by both molecules. For $PIP_2$, a phosphate group (P4 in Fig. 1a) forms salt bridges with both K354 and K358. For CP1, a sulfate group (S2 in Fig. 1a) forms a salt bridge only with K354, whereas K358 interacts with the ring of CP1. However, there are several profound differences between CP1 and $PIP_2$ in their detailed interactions with KCNQ1. First, the sulfate group S2 of CP1 forms a hydrogen bond with residue S253. Second, a phosphate group (P1) of $PIP_2$ and a sulfate group (S1) of CP1 bind to a similar location on KCNQ1 in distinct ways. Specifically, P1 of $PIP_2$ forms a hydrogen bond with residue T247, but S1 of CP1 forms a salt bridge with residue R249. Last, $PIP_2$ contains a third negatively charged group (phosphate group P5) forming a salt bridge with residue R259.

In a previous study, mutation of residues in the predicted $PIP_2$-binding site reduced KCNQ1 currents[18]. CP1 modifies voltage-dependent activation of KCNQ1 channels by shifting the voltage dependence of channel opening, measured as the voltage dependence of the conductance ($G$–$V$) relationship, to more negative voltages (Fig. 1c) among other characteristics (see below). We made mutations to the residues around the CP1-interacting pocket as indicated by molecular docking (Fig. 1b), and examined the effects of the mutations on the CP1-induced shift of the $G$–$V$ relationship. The mutations of the KCNQ1 residues that interact with CP1 in docking simulations reduced the shift of the $G$–$V$ relationship (Fig. 1c, d), supporting the interaction of these residues with CP1.

**CP1 rescues KCNQ1 currents after $PIP_2$ depletion.** We tested whether CP1 can mimic $PIP_2$ in mediating VSD-pore coupling. We co-expressed KCNQ1 with the voltage-dependent lipid

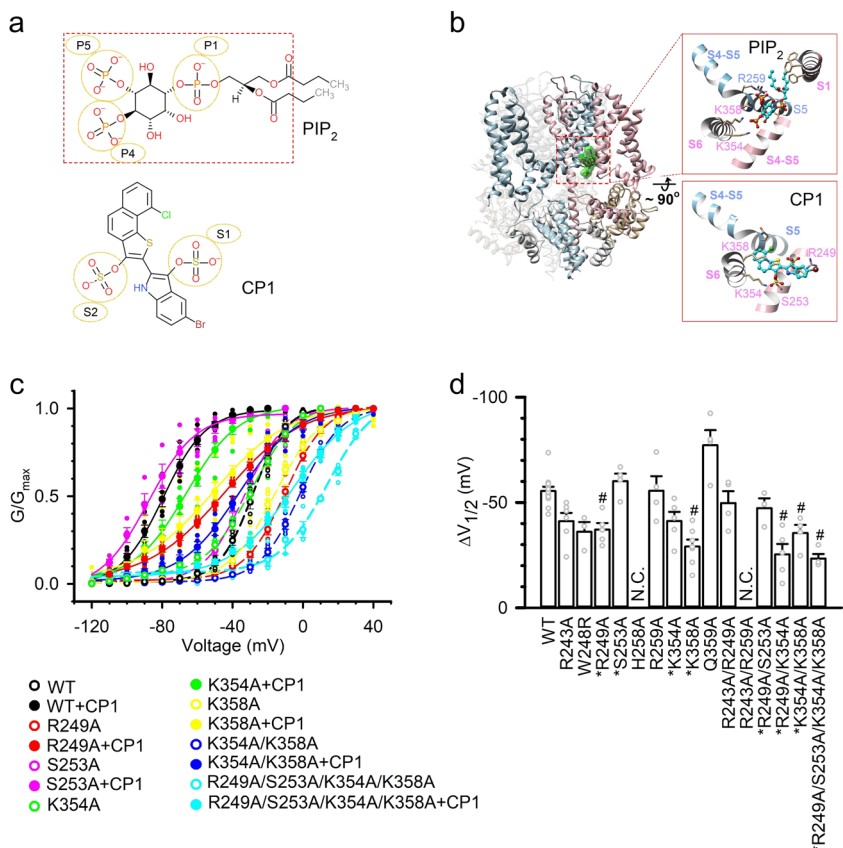

**Fig. 1 A site in KCNQ1 for CP1 interaction. a** $PIP_2$ (upper) and CP1 (lower) molecules. The head of $PIP_2$ that was used for molecular similarity calculations is marked with a rectangle. The negatively charged groups of the two molecules are marked with circles. **b** Left, $PIP_2$ docked on the KCNQ1 and calmodulin complex. Right, magnified view of $PIP_2$ (upper) or CP1 (lower) docked on KCNQ1 at the VSD-PD interface, and the residues interacting with $PIP_2$ or CP1 are indicated. Two neighboring subunits of KCNQ1 are colored sky blue and pink, respectively. The bound calmodulins are colored light gray and tan, respectively. **c** Mutations alter CP1 effects on KCNQ1 conductance–voltage (G–V) relations. The mutated residues are predicted to interact with CP1 in docking. G–V relations of the wild type (WT) and mutant KCNQ1 in the absence (open symbols) or presence (solid symbols) of 10 μM CP1 are shown. **d** The effect of 10 μM CP1 on G–V shift in voltage range of WT and mutant KCNQ1 (WT $n = 11$ and mutant $n = 3$–7). $V_{1/2}$ is the voltage where conductance $G$ is half-maximum, and $\Delta V_{1/2} = V_{1/2}$ (CP1) $- V_{1/2}$ (control). *Residues predicted to interact with CP1 in docking. N.C. no current expression. #Significantly different from the WT ($p \leq 0.05$, Tukey–Kramer ANOVA test). For this and subsequent figures, error bar represents standard error of mean (sem), $n \geq 3$ unless specified otherwise. In this and other experiments, except for those indicated otherwise, CP1 was applied to the bath solution.

phosphatase CiVSP[44] in *Xenopus* oocytes and recorded KCNQ1 currents using two-electrode voltage clamp with consecutive depolarizing voltage pulses. The current increased upon KCNQ1 activation at the beginning of the first voltage pulse (First trace, Fig. 2a) and then declined due to CiVSP activation to deplete $PIP_2$[18]. A subsequent voltage pulse elicited much smaller KCNQ1 currents (Rundown trace, Fig. 2a) as a result of $PIP_2$ depletion that had insufficient time to be replenished by endogenous enzymes between the pulses. However, after application of CP1 via injection into the oocyte, the KCNQ1 currents increased with consecutive voltage pulses, and current kinetics showed no declination during each pulse (10 μM CP1, Fig. 2a, b), indicating that CP1 permits voltage-dependent activation of KCNQ1 channels, despite the depletion of $PIP_2$. Similarly, the $I_{Ks}$ channel (KCNQ1 + KCNE1) currents decrease upon co-expression with CiVSP with consecutive voltage pulses (Fig. 2c, d), which is similar to our previous observations[24]. Application of CP1 via extracellular perfusion rescued $I_{Ks}$ currents, which could be inhibited by the KCNQ1 and $I_{Ks}$ channel inhibitor Chromanol 293B (Fig. 2c, d), indicating that CP1 permits voltage-dependent activation of $I_{Ks}$ channels. This result also shows that CP1 is membrane permeable. To more directly test CP1 effects, we applied CP1 to the intracellular side of the inside-out membrane

patch that expressed $I_{Ks}$ channels. After patch excision in the absence of CP1, the $I_{Ks}$ currents ran down with consecutive voltage pulses due to $PIP_2$ diffusion out from the patch membrane[37], but upon CP1 application, the current increased with consecutive voltage pulses (Fig. 2e, f). These results suggest that similar to $PIP_2$, CP1 association with the KCNQ1 channel can mediate the VSD-pore coupling during voltage-dependent activation.

**CP1 enhances VSD-pore coupling and VSD activation of KCNQ1.** The above results (Fig. 2) are consistent with those from the docking studies (Fig. 1), indicating that CP1 may interact with the channel close to the binding site for $PIP_2$ to mediate VSD-pore coupling. To further understand this mechanism, we examined the changes of KCNQ1 function in the presence of CP1. CP1 modulated the voltage dependence of the KCNQ1 currents by increasing the current at negative voltages ($< -20$ mV), but decreasing the current at positive voltages (Fig. 3a, b), and the voltage dependence of the conductance (G–V) shifted to more negative voltages (Figs. 1c and 3c). Accompanying the negative shift of the G–V relation, the deactivation time course of KCNQ1 channels became slower in CP1 (Fig. 3a, d). These results suggest that CP1 facilitates voltage-dependent activation of

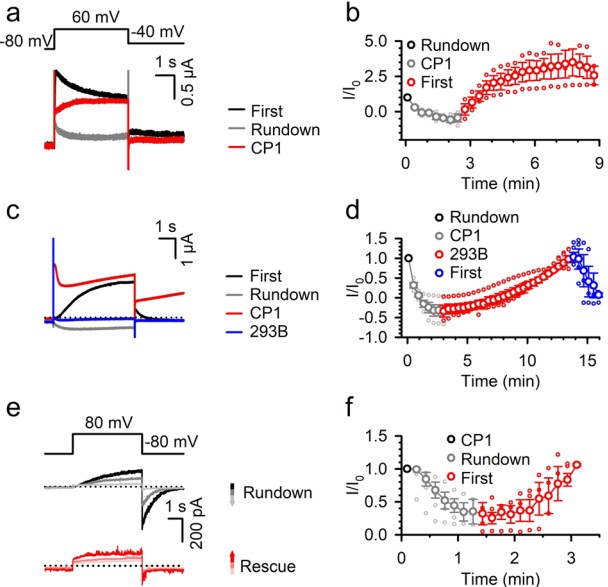

**Fig. 2 CP1 rescues KCNQ1 currents after PIP$_2$ depletion. a, b** KCNQ1 currents recorded from a *Xenopus* oocyte co-expressed with CiVSP in response to voltage pulses to +60 mV (the voltage protocol is depicted in the inset in (**a**). Currents of the first trace control (black), after rundown (gray), and after injection of ~10 μM CP1 into oocytes (red) are shown (**a**). Averaged time course of normalized current amplitude of KCNQ1 with rundown (black) and after CP1 injection (red) (**b**) (n = 3). **c, d** $I_{Ks}$ (KCNQ1+ KCNE1) co-expressed with CiVSP in response to voltage pulses to +60 mV. Currents of the first trace control (black), after rundown (gray), after bath application of 10 μM CP1 (red), and after bath application of 100 μM chromanol 293B (blue) are shown (**c**). Averaged time course of normalized current amplitude of $I_{Ks}$ with rundown (black), CP1 application (red), and chromanol 293B (**d**) (n = 7). **e, f** $I_{Ks}$ currents recorded in the inside-out patch in response to voltage pulses to +80 mV (the voltage protocol is depicted in the inset in (**e**). Representative $I_{Ks}$ current traces ran down after patch excision due to PIP$_2$ depletion (**e**, upper), and rescued by 10 μM CP1 application (**e**, lower). The changing color of the current traces and arrows indicates the time sequence of rundown and rescue (**e**). Normalized current amplitude following patch excision and CP1 application (n = 3) (**f**).

KCNQ1 by favoring pore opening at various voltages. The relation between the *G–V* shift and CP1 concentration is shown in Fig. 3e. The concentration yielding a half-maximum effect (EC$_{50}$) was 8.73 ± 0.68 μM. CP1 does not alter the ion selectivity of KCNQ1 channels (Supplementary Fig. 2).

We employed voltage-clamp fluorometry (VCF) to measure the effects of CP1 on voltage-sensor activation. The emission (*F*) of Alexa 488 C5 maleimide attached to the S3–S4 linker, at residue C219, of the pseudo wild-type KCNQ1 (with mutations C214A/G219C/C331A) reported VSD movements in response to voltage changes, while the ionic current reported pore opening[45] (Fig. 3f, h). Similar to the WT KCNQ1, the *G–V* relation of the pseudo WT KCNQ1 shifted to negative voltages by −53.3 ± 3.1 mV in the presence of 10 μM CP1 (Fig. 3g). CP1 also shifted the *F–V* relation to negative voltages (Fig. 3i), indicating that CP1 potentiates VSD activation. However, the *F–V* relationship shifted only by −17.3 ± 3.6 mV. A larger shift in *G–V* than *F–V* indicates that CP1-enhanced VSD-pore coupling[46,47] (Supplementary Fig. 3). Our results show that a small fraction of VSD activation at negative voltages induces a large fraction of pore opening. It is also apparent that at extreme negative voltages ≤ −130 mV, where the voltage sensor seemed not activated (*F* ~ 0, Fig. 3i), a fraction of the channels was constitutively open (*G* > 0, Fig. 3b, g). These results suggest the idea that CP1 interacts with

the channel, such that pore opening is enhanced even in the absence of VSD activation, and that it mediates VSD-pore coupling during voltage-dependent activation in KCNQ1 channels. A previous study showed a constitutive opening of KCNQ1, which increased when the *G–V* relation was shifted to more negative voltages by mutations[48], but it is not clear if the underlying mechanism is similar to that in CP1 modulation. The above results suggest that while CP1 acts similarly to PIP$_2$ in that it mediates VSD-pore coupling in KCNQ1 channels, its function may differ from that of PIP$_2$, which does not affect VSD activation or opens the pore without VSD activation[18,37].

Our previous studies have shown that the VSD of KCNQ1 activates to an intermediate state (I state) and an activated state (A state) upon membrane depolarization, and the pore can open when VSD is at either the intermediate (IO state) or activated (AO state)[23,24]. The association of the auxiliary subunit KCNE1 with KCNQ1 affects VSD-pore coupling to suppress the IO state and enhance the AO state[24]. Correspondingly, the association of KCNE1 also increases the PIP$_2$ sensitivity of the channel[24,37]. We found that KCNE1 association also affected CP1 modulation of the channel, such that the currents obtained from co-expression of KCNQ1 with KCNE1 exhibited a stronger response to CP1 than KCNQ1 alone. The most prominent effect of CP1 is to dramatically slow the deactivation of $I_{Ks}$, such that at 2 μM CP1, it took a 100-s interval at −120 mV for all the channels to deactivate from their previous activation (Fig. 4a). The application of 2 μM CP1 shifted the *G–V* relation to more negative voltages by ~13 mV (Fig. 4b). If the interval between two testing pulses was shortened to 20 s and held at a less-hyperpolarized voltage of −80 mV, after 2 μM CP1 application, we observed sustained currents, even at −130 mV (Fig. 4c), indicating that a large fraction of the channels was constitutively open at the test voltages. A fraction of the constitutively open channels might have been open during the previous test pulses and not deactivated during the shorter and less-hyperpolarized interval pulse to cause a current accumulation. Nevertheless, we were able to measure the shift of the *G–V* relation under these conditions, which was ~12 mV, a value comparable with that measured in the absence of current accumulation (Fig. 4b, d). In higher concentrations of CP1, we were not able to suppress the constitutive opening of KCNQ1 channels even with long and negative interval voltages, so that we used 20-s and −80-mV interval pulses in the recordings (Fig. 4e). The *G–V* relation showed a large constitutively open component and shifted to even more negative voltages at 10 μM CP1 (Fig. 4f). Current amplitude increased at all voltages, which corresponded with an increase in conductance at all voltages (Fig. 4g, h), suggesting that CP1 increased the maximum open probability of the channel or, alternatively, additional $I_{Ks}$ channels that had been silent in the absence of CP1 were activated by CP1. In comparison with KCNQ1 expressed alone, CP1 caused a larger shift of the *G–V* relationship when KCNE1 was co-expressed (Fig. 4i). Since the $I_{Ks}$ currents were difficult to measure accurately at high CP1 concentrations, we were unable to determine the EC$_{50}$ of the channel response to CP1.

**Specificity of CP1 for KCNQ channels.** KCNQ2 and KCNQ3 form a channel complex that carries M currents in neurons[4]. Similar to KCNQ1, these KCNQ subunits require PIP$_2$ for voltage-dependent activation[49]. KCNQ2 alone can express functional channels in *Xenopus* oocytes, while KCNQ3 alone cannot. However, a single mutation, A315T, allows KCNQ3 to functionally express in *Xenopus* oocytes[50]. We examined the effects of CP1 on KCNQ2, KCNQ3 with mutation A315T (denoted as KCNQ3*), KCNQ2+KCNQ3, and KCNQ2+KCNQ3* expressed in *Xenopus*

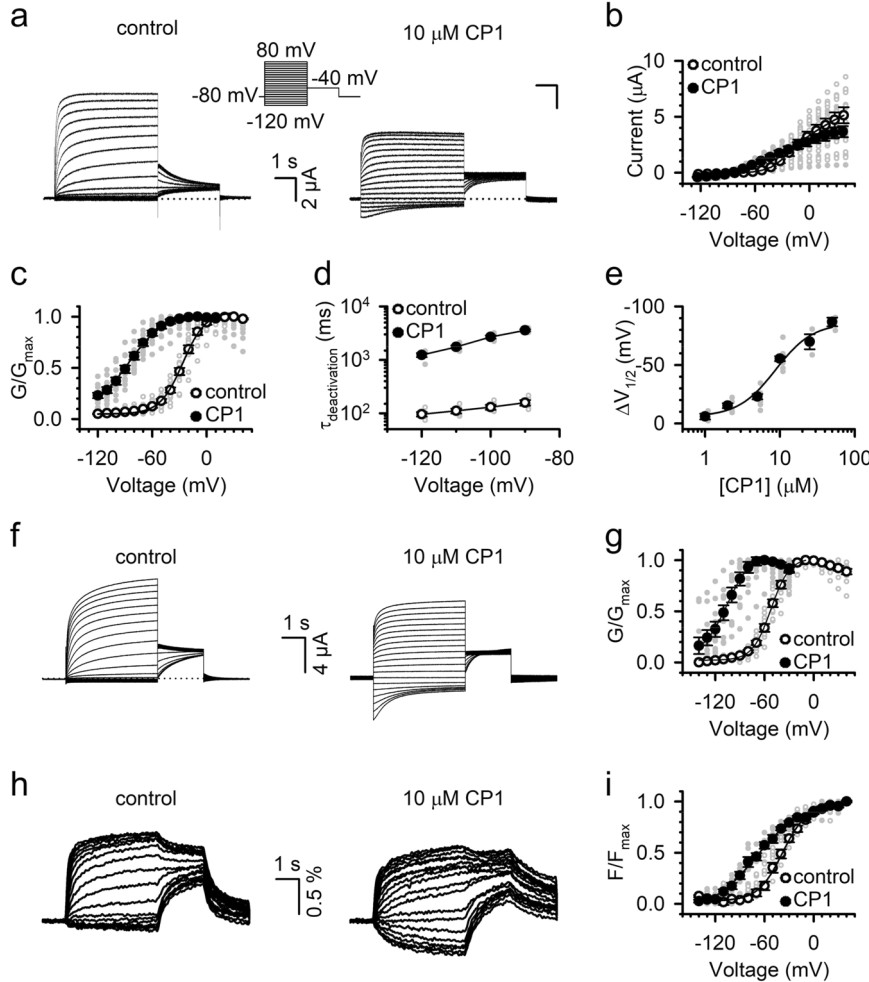

**Fig. 3 CP1 modulates voltage-dependent activation of the KCNQ1 channel. a** KCNQ1 currents elicited in the absence and presence of 10 μM CP1. From a holding potential of −80 mV, test pulses were applied once every 20 s to potentials ranging from −120 mV to +80 mV with 10-mV increments (the voltage protocol is depicted in the inset). The tail currents were elicited at −40 mV. **b** Current–voltage relations of KCNQ1 in the absence or presence of 10 μM CP1. **c** Voltage-dependent activation curves ($G$–$V$) of KCNQ1 in the absence or presence of 10 μM CP1. **d** Time constants of deactivation, obtained by fitting an exponential function to the currents, in the absence or presence of 10 μM CP1. **e** Dose response for mean $\Delta V_{1/2}$ of activation induced by CP1. **f** Current traces of pseudo WT KCNQ1 elicited in the absence and presence of 10 μM CP1. **g** $G$–$V$ curves of pseudo WT KCNQ1 in the absence or presence of 10 μM CP1 ($n = 11$). **h** Fluorescence traces of pseudo WT KCNQ1 in the absence and presence of 10 μM CP1. **i** Fluorescence at steady-state voltage relations of pseudo WT KCNQ1 in the absence or presence of 10 μM CP1 ($n = 7$).

oocytes (Fig. 5a). Similar to the effects on KCNQ1 and $I_{Ks}$ channels, CP1 changed the amplitude of currents (Fig. 5b), shifted the G–V relation to more negative voltages (Fig. 5c), and slowed the deactivation kinetics (Fig. 5d) for all of these KCNQ channels. These CP1 effects were relatively small on KCNQ2 as compared with those on other KCNQ channels, and the effects on KCNQ2/KCNQ3* and KCNQ2/KCNQ3 complexes were closer to those on KCNQ2 alone.

Since CP1 has a similar effect to PIP$_2$ in mediating VSD-pore coupling in KCNQ channels, we tested if CP1 has effects on other ion channels that are also sensitive to PIP$_2$ or are voltage activated and share general structural features of the VSD and the pore with KCNQ. Kir1.1 does not have a voltage sensor, but is activated by PIP$_2$[51], while Ca$_V$1.2, HERG, and HCN4 channels are voltage-gated channels and their function is modulated by PIP$_2$[52–54]. On the other hand, Na$_V$1.5 and K$_V$4.2 are voltage-gated ion channels. In all, 10 μM CP1 showed little effect on the currents of Kir1.1 or G–V relations of all these channels, except for the hyperpolarization-activated HCN4, for which CP1-enhanced currents but shifted the G–V relation to more negative voltages (Fig. 6, Supplementary Fig. 4).

**CP1 reduces drug-induced action potential prolongation.** The KCNQ1 and KCNE1 complex forms the cardiac $I_{Ks}$ channel, which is important in terminating cardiac action potentials and regulating heart rate[1,2,13]. We tested if CP1 modulates the $I_{Ks}$ channel in cardiac myocytes. Similar to the results from KCNQ1 + KCNE1 expressed in *Xenopus* oocytes, in guinea pig ventricular myocyte CP1 enhanced $I_{Ks}$ current (Chromanol 293B sensitive current) amplitude (Fig. 7a–c), shifted the G–V relation to more negative voltages (Fig. 7d, e), and slowed the kinetics of deactivation (Fig. 7a). The response of the amplitude and G–V shift to CP1 doses had EC$_{50}$'s of 7.54 and 7.83 μM, respectively (Fig. 7c, e). The EC$_{50}$ of the G–V shift is slightly smaller compared with that of KCNQ1 expressed in oocytes (8.73 μM, Fig. 3e), possibly because in guinea pig myocytes, KCNQ1 is associated with KCNE1.

In ventricular myocytes, the duration and the morphology of action potentials are determined by various ion channels[55]. Mutations in many of these channels and drugs that modify channel functions, such as an increase in inward Na$^+$ currents or decrease in outward K$^+$ currents, may lead to a prolonged action potential duration, which results in an inherited or acquired long

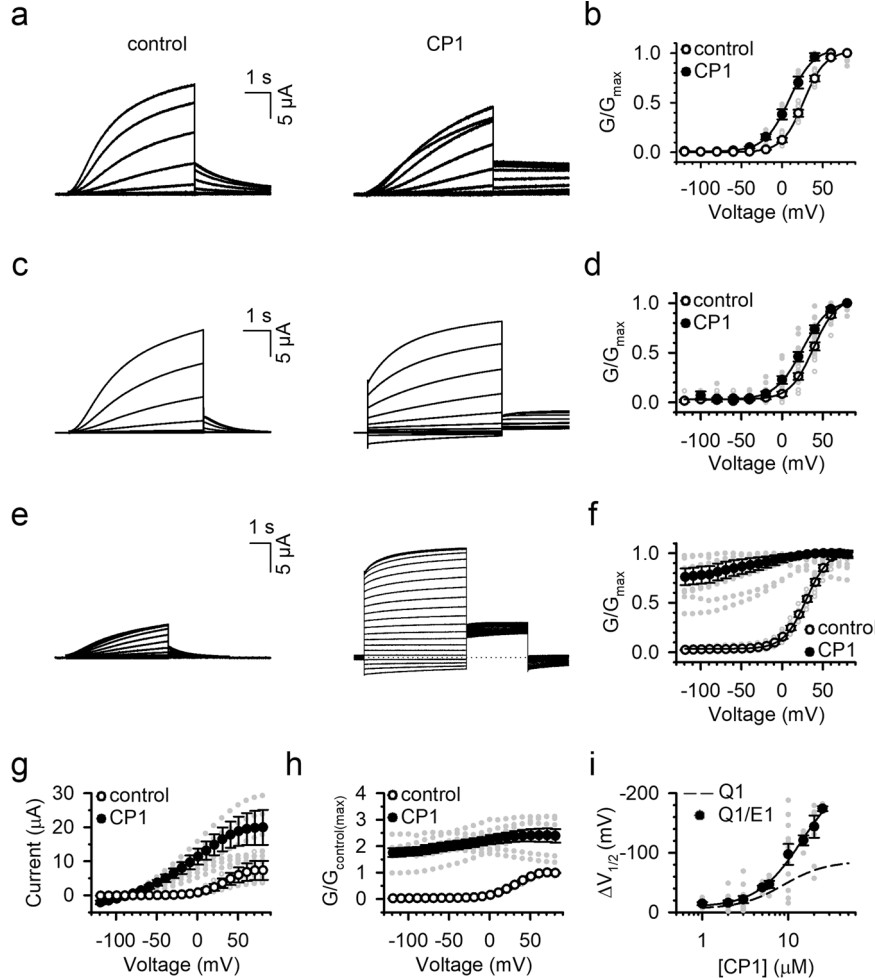

**Fig. 4 Effects of CP1 on $I_{Ks}$ channels. a, b** $I_{Ks}$ currents in the absence (left) and presence of 2 μM CP1 (right) recorded with 100-s and −120-mV interval pulses between testing voltages. Holding potential: −80 mV; testing potential: −120 to +80 mV; returning potential: −40 mV (**a**). G–V relations (**b**). **c, d** $I_{Ks}$ recorded at the same conditions as in (**a**), except for the interval pulses, was 20 s at −80 mV. **e, f** $I_{Ks}$ in the absence or presence of 10 μM CP1. The voltage protocol was the same as in (**b**). **g, h** Current (**g**) and relative conductance (**h**) increases of $I_{Ks}$ channels by 10 μM CP1. In **h**, the measured conductance in CP1 at all voltages was divided by the maximum measured conductance in control, and then normalized with the maximum measured conductance in control being 1. **i** Dose response for $\Delta V_{1/2}$ of G–V relations to different CP1 concentrations as compared with those of KCNQ1 (dashed line).

QT syndrome that predisposes afflicted patients to cardiac arrhythmia[3]. An enhancement of the outward $I_{Ks}$ current may be able to counter these mutations or drug effects, and restore the action potential duration to more normal values. To test this idea, we first perfused the $I_{Ks}$ blocker chromanol 293B together with CP1 to guinea pig ventricular myocytes and found that the action potential duration (APD) was not changed. However, if we continued to apply CP1 alone, the APD was significantly shortened. This indicated that CP1 effects are mediated in large part by enhancing $I_{Ks}$ channels (Fig. 7f). Next, we used an $I_{Kr}$ blocker (Moxifloxacin, Moxi) to prolong action potentials and then applied various concentrations of CP1. We found that 0.2 μM CP1 was sufficient to return the action potential duration back to normal (Fig. 7g, h). Interestingly, 0.2 μM CP1 applied to control myocytes with normal action potentials did not alter action potential duration, suggesting that there is a window of CP1 concentrations that could counter the effects of mutation or drugs that produce prolonged action potentials, but would not alter action potential duration in normal cells. CP1 at concentrations of 0.6 and 6 μM also reduced Moxifloxacin-induced action potential prolongation; however, action potentials in control cells were also shortened (Fig. 7g, h).

## Discussion

Voltage-dependent gating of ion channels involves three fundamental processes: VSD activation, VSD-pore coupling, and pore opening. Ion channels are membrane proteins, and channel functions are regulated by membrane lipids. Previous studies revealed that lipids as a cofactor modulate VSD activation and pore opening in $K_VAP$[56], $K_V2.1$[57], BK[58], and KCNQ $K^+$ channels[31], while in KCNQ1 channels, $PIP_2$ was shown to be required for the VSD-pore coupling[18]. In this study, we found a compound CP1 that resembles the $PIP_2$ head group, and can substitute for $PIP_2$, to mediate VSD-pore coupling in KCNQ channels (Figs. 1 and 2).

Recently, we identified two sets of interactions between the VSD and the pore in KCNQ1 that are important for VSD-pore coupling during voltage-dependent activation[27]. One set of interactions is among residues in the S4–S5L and S6C (Fig. 8) within the same subunit. This set of interactions had been previously identified in other $K_V$ channels[42,43], and was termed as the classic interactions, in which KCNQ1 channels promote channel opening upon VSD movement into the intermediate state, and are also necessary for VSD-pore coupling when VSD is in the activated state. The second set of interactions are among

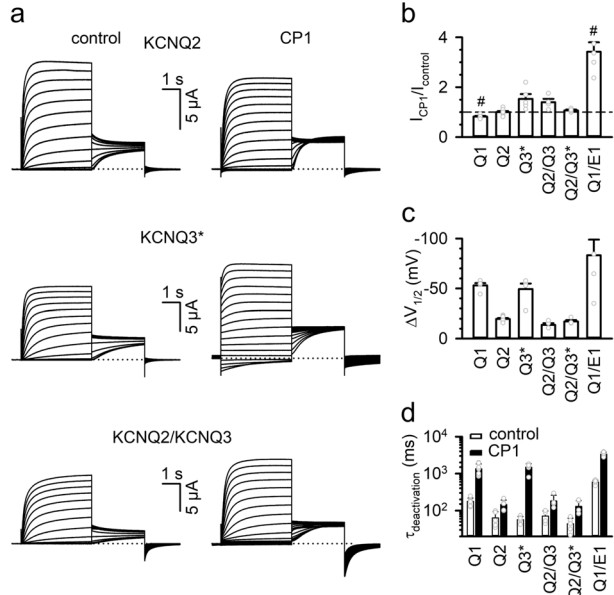

**Fig. 5 Subtype selectivity of CP1 on KCNQ channels. a** Current traces of KCNQ channels as indicated in the absence and the presence of 10 μM CP1. Holding potential: −80 mV, test pulses: −120 to +80 mV with 10-mV step, and returning potential: −40 mV. **b–d** Effects of 10 μM CP1 on the outward current at −10 mV, (KCNQ1/E1 at +40 mV) (**b**), $\Delta V_{1/2}$ (**c**), and deactivation time constant from an exponential fit to the current at −120 mV (**d**) for different KCNQ channels. In **b**, # indicates significant change of current by CP1.

S4, S4–S5L, S5, and S6 between neighboring subunits (Fig. 8), which are engaged by the movement of the VSD to the activated state for VSD-pore coupling[27]. The residues involved in these two sets of interactions for VSD-pore coupling are mapped on the human KCNQ1 structure[59] (PDB entry: 6uzz) along with the residues that interact with PIP₂ and CP1 (see Fig. 1), respectively (Fig. 8). The residues interacting with PIP₂ or CP1 are located within the S4–S5L and S6C, which are right among the residues for VSD-pore interactions (Fig. 8). This result seems to suggest that the connection between S4–S5L and S6C mediated by PIP₂ or CP1 plays an important role in engaging the interactions for VSD-pore coupling. In PIP₂ depletion and the absence of CP1, the interactions for VSD-pore coupling in KCNQ1 may be weakened or disrupted, resulting in the loss of VSD-pore coupling as previously predicted from molecular dynamic simulations[60]. However, it is worth pointing out that PIP₂ interaction with KCNQ channels may be dynamic, which changes with the state of the channel during voltage-dependent activation, to also involve residues in the S2–S3L[18,36,60]. The recently published cryo-EM structure of the hKCNQ1 showed that PIP₂ could bind to S2–S3L[59]. In addition, residues in other structural motifs have been suggested to interact with PIP₂ during KCNQ channel activation[33,61,62]. It is not known if CP1 makes dynamic interactions with any other sites as well. It is possible that some of the effects of CP1, such as causing a constitutive activation in the absence of VSD activation, may derive from CP1 binding to a different binding site. It has been shown that polyunsaturated fatty acids modify KCNQ1 channels by interacting with residues in both the VSD and S6[31].

While PIP₂ is required for the activation of all KCNQ channels, the properties of each of the KCNQ channels may differ under PIP₂ modulation. In response to PIP₂ applied to intracellular solutions, KCNQ3 channels activate in response to increased PIP₂

concentrations with an ~100-fold smaller EC₅₀ (higher apparent affinity) than KCNQ2 or KCNQ4[63]. Consistently, the sites for PIP₂ and CP1 interaction are generally conserved but show some differences (Supplementary Fig. 5). On the other hand, KCNQ1 activation showed an EC₅₀ more than 100-fold higher in response to PIP₂, than the co-expression of KCNQ1+KCNE1[37]. For KCNQ1+KCNE1 channels, PIP₂ not only increases activation but also shifts the $G–V$ relation to more negative voltages[37,64]. However, following muscarinic stimulation to partially deplete PIP₂ or enzymatic treatment that altered PIP₂ levels, KCNQ1, KCNQ2, KCNQ4, or KCNQ2/KCNQ3 channels did not show a shift in $G–V$ relations[65–67]. Similar to PIP₂, CP1 also modulates different KCNQ channels with differing $G–V$ shifts; in response to 10 μM CP1, the $G–V$ shift ranks in the order KCNQ1 + KCNE1 > KCNQ1 ~ KCNQ3* > KCNQ2 ~ KCNQ2/KCNQ3 (Figs. 3–5). However, the effects of CP1 on KCNQ1 channels also show distinct differences from PIP₂. PIP₂ depletion does not alter VSD activation in KCNQ1[18], but CP1 shifts VSD activation to more negative voltages (Fig. 3). CP1 also shifts the $G–V$ relation to more negative voltages more prominently and causes a constitutive current even when the VSD is at rest at extremely negative voltages (Figs. 3 and 4). These results suggest that the VSD-pore-coupling mechanism in KCNQ1 is distorted by an interaction with CP1, which not only allows pore opening with the VSD at rest, but also alters the voltage dependence of VSD activation. In our previous studies, we have found that single mutations of many individual amino acids[23–25,27,68] and the depletion of PIP₂[18] could abolish VSD-pore coupling completely. This study suggests that, while the interaction of CP1 could restore VSD-pore coupling in the absence of PIP₂ (Fig. 2), it also distorts VSD-pore coupling to alter activation properties (Figs. 3, 4) due to the different interactions of CP1 and PIP₂ with the channel protein (Figs. 1, 8). All these results suggest that the VSD and pore in KCNQ1 channels have a coupling that is prone to modulation by structural disturbances.

In all, 10 μM CP1 shows no effects on voltage-dependent activation of some ion channels other than KCNQ that are voltage dependent, PIP₂ sensitive, or both (Fig. 6), supporting the idea that its effects on KCNQ channels are site specific but not due to a nonspecific electrostatic interaction with the channels (Supplementary Fig. 5). Previous studies suggested that the length of the fatty acid chains of PIP₂ may not contribute to the activation of KCNQ1 channels[37]. However, in a systematic study, Brown and colleagues found that a minimum of one acyl chain was required for inositol phosphates to activate the KCNQ2/KCNQ3 channel. The water-soluble inositol head group I(1,4,5) P₃, I(4,5)P₂, or other small phosphates had no effect on channel activity. On the other hand, any phosphate head group of a lipid could activate the channel, with PIP₂ being the most effective[69]. Taken together with our results with CP1, it seems that a lipophilic moiety may be necessary to anchor the inositol head group for interaction with and activation of the channel, but any negatively charged groups mimicking the phosphate groups that can bind to the channel, such as CP1, will be able to activate the channel without the participation of an acyl chain. Alternatively, since externally applied CP1 could penetrate the membrane to interact with the site in the cytosolic domain of the channel (Figs. 1–4), CP1 may interact with the membrane while interacting with the channel. It is worth noting that all lipid phosphates examined in the previous study did not appear to alter channel activation, except for changing the maximal channel activity[69], while CP1 interaction changes VSD activation and VSD-pore coupling (Figs. 3 and 4). It indicates that while phosphates are important, the structural differences between CP1 and lipid head groups permit CP1 to interact with other residues and cause additional functional impacts.

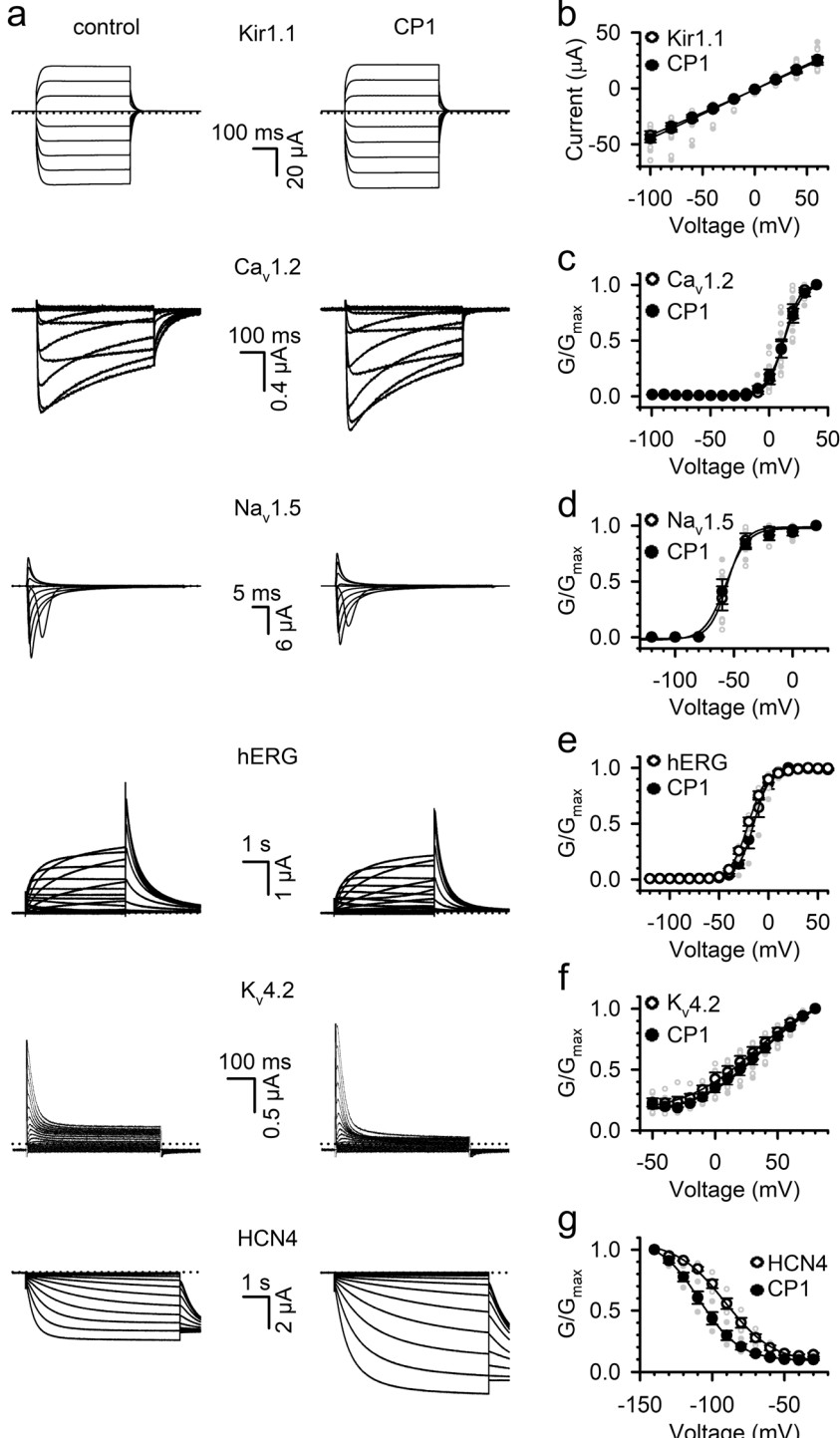

**Fig. 6 Effects of CP1 on other ion channels expressed in *Xenopus* oocytes. a** Currents of Kir1.1, $Ca_V1.2$, $Na_V1.5$, hERG, and $K_V4.2$ before and after application of 10 μM CP1, respectively. The voltages for holding, test, and returning pulses were Kir1.1, 0 mV, −100 to +60 mV, 0; $Ca_V1.2$, −100 mV, −100 to +60, −100 mV; $Na_V1.5$, −100, −120 to +40 mV, −120 mV; hERG, −80 mV, −90 to +60 mV, −60 mV; $K_V4.2$, −80 mV, −100 mV to +80 mV, −40 mV; HCN4, −30 mV, −30 to −140 mV, −120 mV. **b** Current–voltage relations of Kir1.1 channel in the absence or presence of 10 μM CP1 ($n = 7$). **c–g** $G$-$V$ relations of indicated channels with and without 10 μM CP1. The $\Delta V_{1/2}$ of $G$-$V$ relations (mV) are $Ca_V1.2$, 3.38 ± 0.71 mV (**c**); $Na_V1.5$, −2.26 ± 0.95 mV (**d**); hERG, 6.85 ± 1.81 mV (**e**); $K_V 4.2$, −2.26 ± 2.88 mV (**f**); HCN4, 17.13 ± 1.59 mV (**g**).

KCNQ1 harbors more than 300 loss-of-function mutations that reduce $I_{Ks}$ currents and are associated with long QT syndrome (LQTS). We found that CP1 application to ventricular myocytes increased $I_{Ks}$ currents and reduced action potential duration (Fig. 7), suggesting a potential for anti-arrhythmic therapy. CP1 at low concentration (0.2 μM) reversed drug-induced action potential prolongation, but showed no effect of its own on normal action potentials, suggesting a window between therapeutic effects and cardiac toxicity. CP1 also showed good specificity for $I_{Ks}$ as compared with the neuronal M currents (KCNQ2 and KCNQ2/KCNQ3, Fig. 5) and other important cardiac ion channels (Figs. 6 and 7). These results suggest that our

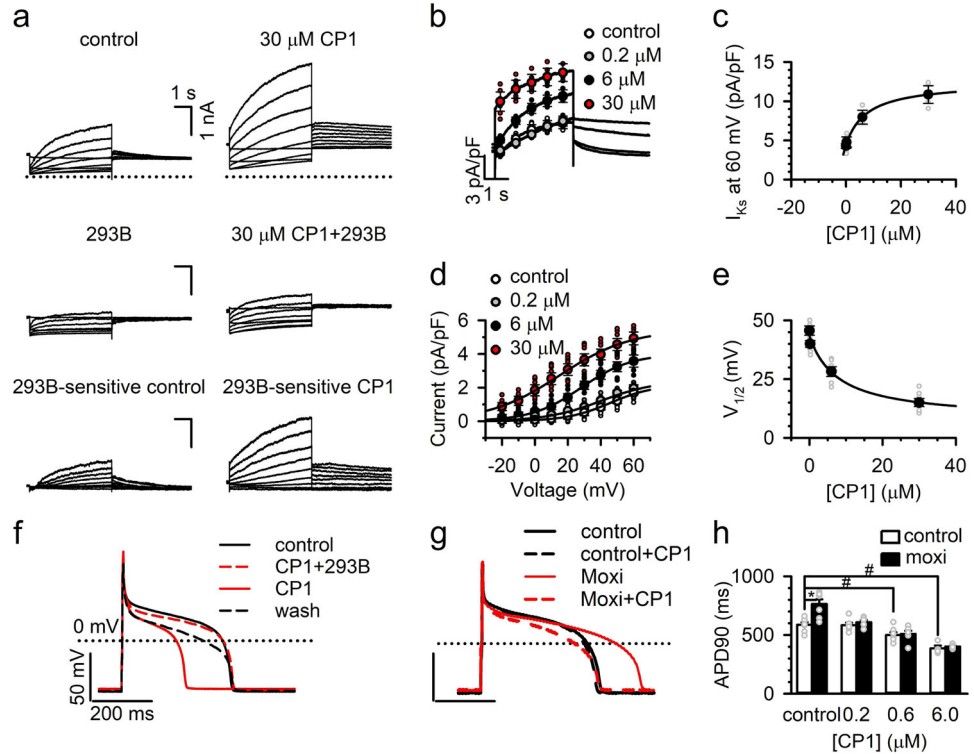

**Fig. 7 Effects of CP1 on $I_{Ks}$ and action potentials in cardiomyocytes. a** $I_{Ks}$ currents in guinea pig ventricular myocytes in the absence and presence of 30 μM CP1 in the whole-cell patch-clamp configuration. Holding potential: −40 mV; testing potentials: −20 to +60 mV with 10-mV increment; returning potential: −20 mV. The $I_{Ks}$ and its tail currents at the returning potential in control were obtained by subtracting the currents in the presence of chromanol 293B (10 μM) from those in the control only, and the $I_{Ks}$ and its tail currents in the presence of CP1 were obtained by subtracting the currents in the presence of CPI (30 μM) plus 293B (10 μM) from those in the CPI only. **b** Averaged $I_{Ks}$ currents in control and different CP1 concentrations [CP1] at 60 mV. **c** Dose response of $I_{Ks}$ channels at +60 mV for CP1, $EC_{50} = 7.54$ μM. **d** Averaged $I_{Ks}$ tail currents in control and different [CP1] at −20 mV. **e** Dose response for $V_{1/2}$ of activation induced by CP1 for $I_{Ks}$ channels, $EC_{50} = 7.86$ μM ($n = 6$). **f** Effects of CP1 on normal actional potential duration (APD). Guinea pig ventricular myocytes were first perfused with 10 μM CP1 and 10 μM chromanol. After APD reached steady state, 10 μM CP1 was constitutively perfused alone. Last, CP1 was washed out for near-full reversal of APD shortening. **g** Effects of 0.2 μM CP1 on LQT action potentials. To mimic the LQT, 100 μM moxifloxacin was applied 2 h before the treatment of CP1. **h** Change of action potential duration after application of different [CP1]s ($n = 5$–7). Tukey–Kramer ANOVA test was used to compare control cells in different CP1 concentration, # is significant at $P < 0.05$. Unpaired two-tailed Student $t$ tests were used to compare control and moxifloxacin cells at different CP1 concentration: * is significant at $P < 0.05$.

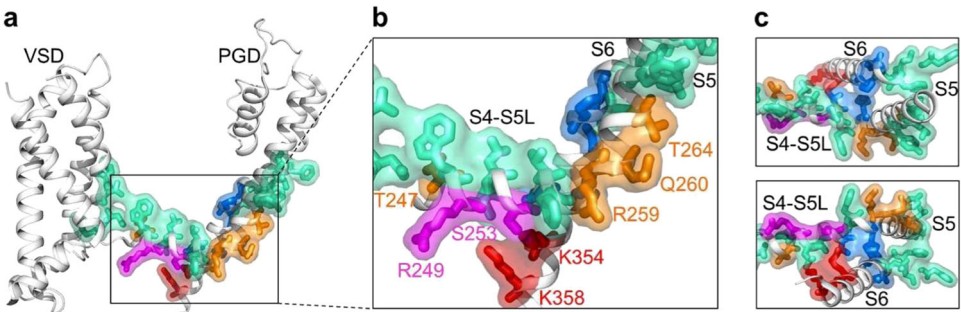

**Fig. 8 Residues important for VSD-pore coupling in KCNQ1.** The residues are shown as colored sticks in the cryo-EM structure of hKCNQ1 (PDB entry: 6uzz). The colors indicate residues in the classic interactions (blue, including V254, H258, A341, P343, and G345), the interactions specifically when the VSD is at the activated state (cyan, including M238, L239, D242, R243, W248, L250, L251, V255, F256, Y267, I268, L271, G272, F335, S338, F339, and L342), specific interactions with $PIP_2$ (orange, T247, R259, Q260, and T264), specific interactions with CP1 (magenta, R249 and S253), and interactions with both $PIP_2$ and CP1 (red, K354 and K358). **a** One KCNQ1 subunit. **b** Enlarged frame. **c** Enlarged frame from different views.

computational strategy to identify CP1 based on the structural data of $K_V$ channels, and the understanding of $PIP_2$ interactions with KCNQ channels, can be effective as an approach for drug discovery targeting ion channels. This strategy may have general applications with ever more readily available structural and functional data of ion channels.

## Methods

**Homology models of hKCNQ1 and in silico compound screening.** A hybrid in silico screening strategy, combining both structure-based and ligand-based methods, was used in this study. For structure-based screening, by using an in-house molecular docking software MDock[39,70,71], we screened a subset of the Available Chemical Database (ACD, Molecular Design Ltd.) in which each compound has a formal charge of 2 (~$10^4$ compounds), targeting the $PIP_2$ site (green in Fig. 1b) on

human KCNQ1 (hKCNQ1). The hKCNQ1 structure was constructed based on the crystal structure of rat $K_V1.2$–$K_V2.1$ chimera (PDB entry: 2r9r)[72] using the program MODELLER[73]. For ligand-based screening, by using a 3D molecular similarity calculation program ShaEP[41], we ranked the compounds in the subset of ACD by their similarity scores with respect to the head group of $PIP_2$. The best consensus compound, CP1, ranked in the top 2% for both structure-based screening (#197) and ligand-based screening (#34), was selected for experimental studies.

In the interim of the study, the structure of the *Xenopus* KCNQ1 channel (fKCNQ1) was solved using cryo-EM and published (PDB entry: 5vms)[22]. We then built a second homology model of hKCNQ1 based on the structure of fKCNQ1 (ribbons in Fig. 1b). This homology model was used to study the detailed binding mode of $PIP_2$ and CP1 in KCNQ1. $PIP_2$ and CP1 were docked using the homology model of hKCNQ1 built upon 5vms. The predicted binding modes of $PIP_2$ and CP1 on hKCNQ1 are shown in Fig. 1b, plotted with Chimera[74]. Very recently, a human KCNQ1 structure was released using cryo-EM (PDB code: 6uzz; resolution: 3.1 Å)[59]. Our modeled hKCNQ1 structure based on fKCNQ1 (PDB code: 5vms) was very close to this cryo-EM hKCNQ1 structure, with the backbone root-mean-square deviation (RMSD) of 1.5 Å. Hence, the use of the newly released cryo-EM human KCNQ1 structure for modeling does not change our docking results.

**Channel subunit and mutation cRNA preparation.** Complementary DNA (cDNA) encoding human KCNQ1 (UniProtKB/Swiss-Prot under accession no. P51787), KCNQ2 (O43526), KCNQ3 (O43525), KCNE1 (P15382), Kir1.1 (P48048), hERG (Q12809), $K_V4.2$ (Q63881), $Na_V1.5$ (Q14524), $Na_V$ β1 (Q07699), $Ca_V1.2$ (Q13936), $Ca_V$ β1a (Q02641), $Ca_V$ α2/δ1 (P54289), and HCN4 (Q9Y3Q4) were subcloned into oocyte expression vectors, respectively. Site-directed mutations of the KCNQ1 channel were all produced by overlap extension polymerase chain reaction (PCR) with high-fidelity *Pfu* polymerase (Stratagene, CA). The presence of the desired mutation was verified by sequencing the PCR-amplified regions. All the primer sequences used in this study are available in Supplementary Table 1. The complementary RNA (cRNA) of all these channels and mutations was transcribed in vitro with the T3, T7, or SP6 polymerase mMessage mMachine kit (Invitrogen, Thermo Fisher Scientific, MO).

**Oocyte isolation and cRNA injection.** The use of *Xenopus laevis* to obtain oocytes was approved by the Washington University Animal Studies Committee (protocol #20190030). Oocytes were manually dispersed and digested with collagenase (0.5 mg/ml, Sigma-Aldrich, Saint Louis, MO) to remove the follicle cell layer. Stage V or VI oocytes were selected and injected with cRNAs (9.2 ng) of channel or mutations. Especially, to characterize $I_{Ks}$, a mix of cRNAs at a molar ratio 4:1 (KCNQ1: KCNE1) was injected. For KCNQ2/KCNQ3, the molar ratio was 1:1 (KCNQ2: KCNQ3). For sodium channels, a 3:1 molar ratio (β1:$Na_V1.5$) and for calcium channels a weight ratio 1:1:1 (β1:α2/δ1:$Ca_V1.2$) was injected, respectively. Injected oocytes were incubated in ND96 solution containing (in mM) 96 NaCl, 2 KCl, 1.8 $CaCl_2$, 1 $MgCl_2$, and 5 HEPES, pH 7.6, supplemented with 2.5 mM Na pyruvate and 1% penicillin–streptomycin, at 18 °C for 3–7 days before recording.

**Cardiac myocyte isolation.** The use of guinea pigs was approved by the Stony Brook University Institutional Animal Care and Use Committee. Single ventricular myocytes were acutely enzymatically isolated from guinea pig heart as previously described[75]. Guinea pigs, weighing 300–500 g, were sacrificed by peritoneal injection of sodium pentobarbitone (1 ml, 390 mg/ml). The isolated cells were stored in KB solution containing (in mM) 83 KCl, 30 $K_2HPO_4$, 5 $MgSO_4$, 5 Na pyruvate, 5 β-OH-butyric acid, 20 creatine, 20 taurine, 10 glucose, and 0.5 EGTA, pH 7.2.

**Electrophysiology.** All electrophysiological recordings on oocytes and isolated ventricular cells were performed at room temperature (20–23 °C).

*Two-electrode voltage clamp.* The whole-oocyte ionic currents were recorded with a GeneClamp 500B amplifier (Axon Instruments, CA) or iTEV90 amplifier (HEKA, Germany) and then sampled at 1 kHz and low-pass-filtered at 2 kHz. The Patchmaster software (HEKA, Germany) was used to acquire data. The Kir1.1 current was recorded with a high $K^+$ bath solution containing (in mM) 96 KCl, 4 NaCl, 1 $CaCl_2$, 2 $MgCl_2$, and 10 HEPES, pH 7.3–7.4. The HCN4 current was recorded in ND66 solution containing (in mM) 66 NaCl, 32 KCl, 1.8 $CaCl_2$, 1 $MgCl_2$, and 5 HEPES, pH 7.6. All the other potassium channels were recorded in ND96 solution. The currents of $Ca_V1.2$ channel were recorded in 40 mM $Ba^{2+}$ bath solution containing (in mM) 40 Ba(OH)$_2$, 50 TEA-OH, 2 KOH, and 5 HEPES, adjusted to pH 7.4 with methanesulfonic acid. CP1 (DISODIUM 5-BROMO-2-[9-CHLORO-3-(SULFONATOOXY) NAPHTHO [1,2-B] THIEN-2-YL]-1H-INDOL-3-YL SULFATE, Sigma-Aldrich, MO) and Chromanol 293B (Sigma-Aldrich, MO) were dissolved in bath solutions. All the chemicals were purchased from Sigma-Aldrich. To measure the $I$–$V$ relations of the Kir1.1 channel, the test voltage was set from −100 to +60 mV for 1 s with 20-mV increments. The test pulses were stepped from a holding potential of 0 mV and then stepped back to 0 mV. The HCN4 currents were recorded by a test pulse between −30 and −140 mV in 10-mV step from a holding potential of −30 mV. For all the other potassium channels and $Ca_V1.2$ channel, the holding potential was −80 or −100 mV, and the test pulse was

applied with 10-mV increment, which was followed with a repolarization pulse at −40 mV before returning to the holding potential. The durations of test and the repolarization pulse vary for different channels to allow the activation and deactivation of the channel to reach their steady states.

*Cut-open oocyte recording.* $Na_V1.5$ currents were recorded with a cut-open amplifier (CA-1B, Dagan Corporation) coupled with an A/D converter (Digidata 1440, Molecular Devices). The data were acquired using Clampex software (v10, Molecular Devices). The internal solution was composed of (mM) 105 NMG-Mes, 10 Na-Mes, 20 HEPES, and 2 EGTA, pH 7.4. The external solution was composed of (mM) 25 NMG-Mes, 90 Na-Mes, 20 HEPES, and 2 Ca-Mes$_2$, pH 7.4. The ionic currents were recorded using the standard $I$–$V$ protocol. From a holding potential of −120 mV, cells were stepped to a 100-ms prepulse of −120 mV and then stepped to test potentials ranging from −120 to 60 mV with 20-mV increment, preceded by a 100-ms postpulse of −120 mV.

*Voltage-clamp fluorometry.* Oocytes were labeled on ice for 45 min with 10 μM Alexa 488 C5 maleimide (Molecular Probes, Eugene, OR) in high $K^+$ solution (in mM: 98 KCl, 1.8 $CaCl_2$, and 5 HEPES, pH 7.6). Then the cells were washed with ND96 solution and kept on ice until recording. A CA-1B amplifier was used to record whole-oocyte currents in ND96 solution. Fluorescent signals were recorded simultaneously using a Pin20A photodiode (OSI Optoelectronics, CA), filtered using a FITC filter cube (Leica, Germany, for Alexa 488), and then amplified using a patch-clamp amplifier (EPC10, HEKA, Germany).

*Inside-out patch clamp.* Macroscopic currents expressed in *Xenopus* oocytes were recorded from inside-out patches formed with borosilicate pipettes of 0.5–1.0-MΩ resistance. The data were acquired using an Axopatch 200-B patch-clamp amplifier (Axon Instruments) and Pulse acquisition software (HEKA). The pipette solution contained (in mM) 140 $KMeSO_3$, 20 HEPES, 2 KCl, and 2 $MgCl_2$, pH 7.2. The internal solution contained (in mM) 140 $KMeSO_3$, 20 HEPES, 2 KCl, and 5 ethylene glycol tetraacetic acid (EGTA), pH 7.2.

**$I_{Ks}$ measurements in isolated ventricular cells.** An Axopatch 1D amplifier (Axon Instruments, Inc.) and the whole-cell patch-clamp technique were used to measure the $I_{Ks}$ current. The patch pipette solution contained (in mM) 140 $KMeSO_3$, 20 HEPES, 2 KCl, 5 EGTA, and 5 MgATP, pH 7.4. The external Tyrode contains (in mM) 137.7 NaCl, 2.3 NaOH, 8 KCl, 1 $MgCl_2$, 5 HEPES, 1 $CdCl_2$, and 10 Glucose, pH 7.4. The cells were held at −40 mV and stepped to −20 mV, then to +60 mV with an increment of 10 mV, to measure the membrane current in the absence and the presence of chromanol 293B (10 μM). The chromanol 293B sensitive current was defined as the $I_{Ks}$ current normalized to its cell capacitance. These experimental protocols and conditions eliminate most of the other membrane currents, such as $I_P$, $I_{Na}$, $I_{Ca}$, and $I_{Na/Ca}$, except $I_K$, so that we could obtain better experiment resolution.

**Electrophysiological recording of action potentials.** Action potentials were recorded with whole-cell patch-clamp recording techniques. Current-clamp configuration was used. Freshly isolated guinea pig cardiac myocytes were placed at 1 Hz with a stimulus of 180-pA amplitude and 10-mS duration. The APD was determined at 90% repolarization. Several minutes were allowed for the APD to reach steady state before data were collected. Typical Tyrode mimic AP Solution for action potential recording was used. The bath solution is composed of (in mM) 140 NaCl, 3 KCl, 1 $MgCl_2$, 1.8 $CaCl_2$, 10 HEPES, and 10 Glucose, pH 7.4. The pipette solution is composed of (in mM) 115 K-aspartic acid, 35 KOH, 3 $MgCl_2$, 10 HEPES, 11 EGTA, 5 Glucose, and 3 MgATP, pH 7.4.

**Kinetic modeling.** Our previous studies[23,24,27] proposed a five-state Markov model to conceptually recapitulate the gating process of KCNQ1 channels that involve two steps of VSD activation and two open states. We used this model to illustrate, in principle, that the CP1 effects on KCNQ1 channel gating can be produced by enhancing the VSD-pore coupling (Supplementary Fig 3). In this model, resting closed, intermediate closed (IC), and activated closed (AC) stand for VSD conformations at resting, intermediate, and activated states when the pore is closed, IO and AO are the pore opening at intermediate and activated states. Different states are connected by transition rates, where $\alpha_1 = a_1 \times \exp(v/m)$, $\beta_1 = c_1 \times \exp(-v/n)$, $\alpha_i = a_i \times \exp(v/b)$, and $\beta_i = c_i \times \exp(-v/d)$ ($i = 2$ and 3) are voltage-dependent transitions for the VSD activation, and k1–4 are rates (constant) for closed–open transitions, which describe the VSD-pore coupling. The values of the parameters in the simulations (Supplementary Fig 3) are as follows: $a_1 = 0.00070$ ms$^{-1}$, $a_2 = 0.0047$ ms$^{-1}$, $a_3 = 0.15$ ms$^{-1}$, $c_1 = 0.0020$ ms$^{-1}$, $c_2 = 0.00017$ ms$^{-1}$, $c_3 = 0.030$ ms$^{-1}$, $m = 46.0$ mV, $n = 31.2$ mV, $b = 37.7$ mV, $d = 41.5$ mV, $k_1 = 1.5$ (for control), and $k_1 = 2$ (for +CP1), $k_2 = 853.08$, $k_3 = 1$ (for control), $k_3 = 4$ (for +CP1), and $k_4 = 103.82$. $c_3$ was set to 0.090 ms$^{-1}$ to balance the model. The output states for VSD movements are the combination of IC, IO, AC, and AO, and the output states for currents are the combination of IO and AO. Note that this is a simplified model without considering that one KCNQ1 channel is formed by four subunits. This model and the experimental data are not sufficient to describe the detailed mechanism of voltage-dependent gating or CP1 modulation of KCNQ1 channels.

**Statistics and reproducibility**. Electrophysiology data were processed with IGOR (Wavemetrics, Lake Oswego, OR), Clampfit (Molecular Devices, Sunnyvale, CA), and SigmaPlot (SPSS, Inc., San Jose, CA). Normalized tail currents were plotted versus prepulse voltage and fitted with the Boltzmann function $G = G_0 + 1/(1 + \exp(V - V_{1/2})/S)$, where $G_0$ is the minimum conductance, $V_{1/2}$ is the half-maximal voltage of activation, and $S$ is the slope factor. Dose–response curves in Fig. 3 were fitted with the Hill equation, $E = E_{max}/(1 - EC_{50}/C)^P$, where $EC_{50}$ is the drug concentration producing the half-maximum response and $P$ is the Hill coefficient. In Fig. 7c, e, the $EC_{50}$ values were obtained by fitting the data to $I_{Ks} = I_{Min} + (I_{Max} - I_{Min})$ ([CP1]/([CP1] + $EC_{50}$)) and $V_{1/2} = V_{1/2 min} + (V_{1/2Max} - V_{1/2 min})$ ([CP1]/([CP1] + $EC_{50}$)). The deactivation time course was fitted with single-exponential function. All data are expressed as mean ± SEM ($n \geq 3$ or otherwise indicated). Electrophysiology experiments were performed on at least two separate batches of oocytes or myocytes to confirm reproducibility. The significance was estimated either using unpaired two-tailed Student $t$ tests (Fig. 7g) or one-way Tukey–Kramer ANOVA test (Fig. 1d and Fig. 7g). Statistical significance: $P \leq 0.05$.

**Reporting summary**. Further information on research design is available in the Nature Research Reporting Summary linked to this article.

## Data availability
Raw data used to generate the charts present in this paper can be found in Supplementary Data 1. Additional data and research materials related to this paper are available from the corresponding author on reasonable request.

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

## Acknowledgements
This work was supported by R01 HL126774 to J.C., by R01 GM109980, and R35 GM136409 to X.Z., and by AHA 18POST34030203 to P.H.

## Author contributions
J.C., X.Z., and I.C. conceived the study. X.X. and X.Z. performed the in silico screening and docking studies. Y.L., M.M.N., H.L., W.Z., and P.H. performed the voltage-clamp recording. Y.L. and N.-D.Y. performed the VCF recording. H.L. and W.Z. performed the inside-out patch-clamp recording. J.G. and H.Z.W. performed the electrophysiology work with isolated guinea pig myocytes. J.S., K.M.W., Y.L., W.K., A.D., and A.C. performed the molecular biology. P.H. performed the kinetic modeling. All authors contributed to data analysis. Y.L., J.C., X.Z., I.C., X.X., and G.Z. wrote the paper with input from all authors.

## Competing interests
J.S. and J.C. are cofounders of a startup company VivoCor LLC, which is targeting $I_{Ks}$ for the treatment of cardiac arrhythmia. Other authors declare they have no competing interests.
