## [Peer Review File · Communications Biology]

Reviewers' comments:

Reviewer #1 (Remarks to the Author):

The authors here identify CP1 as a substitute for PIP2 mediating the VSD-pore coupling in KCNQ1 channels. CP1 rescued KCNQ1 currents when membrane PIP2 was depleted. Similar to PIP2, CP1 interacts with several amino acids that have been shown to be very important for VSD-pore coupling. However, compare to PIP2 that does not affect the VSD activation or pore opening in the absence of VSD activation, CP1 enhanced the VSD-pore coupling and pore opening in the absence of VSD activation of KCNQ1 channel. Also, the authors found that CP1 had a similar effect on mediating VSD-pore coupling in all KCNQ channels including changing the current amplitude and shifting the G-V relation to more negative voltages. But CP1 showed no effects on other PIP2-sensitive ion channels such as Kir1.1, Cav1.2 and HERG channels. In addition, CP1 shortened the action potential prolongation caused by drug-induced prolongation of ventricular myocytes. By testing a compound CP1 that can mediate the voltage dependent activation of KCNQ1 channels and modulate the drug-induced action potential in ventricular myocytes, the authors conclude that this work helps to develop a potential approach for anti-arrhythmic treatment. The paper is well-organized and easy to read. The experimental data seem convincing and the conclusion seems reasonable. It would attract readers who are interested in the VSD-pore coupling of voltage-gated ion channels and in developing new compounds for the therapy of channelopathies. However, I would like some more discussion with alternative explanations, as listed below.

Major comments:

1. Line 167-169. I don't think these papers are saying that a large fraction of channels be open when only small fraction of channels have moved their S4? How do you explain a FV at more positive voltages than the GV? At least provide a model simulation that gives this phenomena.
2. Fig 2f. Why the run down kinetics seems much faster than it in a previous study (Li et.al, 2011)?
3. Line 171, is the constitutively conducting current K current? Could it be endogenous current or other currents?
4. Line 171-174. I think this study shows the opposite, increasing the Gmin of the pore shifts the GV by allowing relatively more S4 movement in open channels instead of in closed channels. By microreversibility, the S4 movement happens in open channels at more hyperpolarized voltages than in closed channels. I have to point out that the equation given in the paper by Ma et al. is incorrect and due to faulty mathematical rearrangements of the equations. There is no dependence of Gmin on $V_{0.5}$: $G_{min} = 1/(1+K)$, where K is the equilibrium constant between C and O with S4 down.
5. Line 179. Could CP1 have effects on the pore independent of coupling? This should at least be entertained.
6. Line 187. "coexpression of KCNQ1 with KCNE1 exhibited a stronger response to CP1 than KCNQ1 alone." Is this really true.? Seems like voltage shift is bigger in KCNQ1 alone? How do you quantify stronger response? Maybe this is just due to your inability to hold negative enough to close completely channels between pulses? Some quantitative effect on a channel parameter would be best to show, i.e. Delta $V_{0.5}$, G_{max} increase, etc..
7. Line 193, if after treating with 2 μ M CP1, the IKs channels show a big fraction of constitutive currents at the very negative voltages, why it is not reflected in the GV relation in Fig 4d.
8. Line 204. "suggesting additional IKs channels that had been silent in the absence of CP1 were activated by CP1.". Maybe just a change in maximum open probability?
9. A VCF experiment showing the effects of CP1 on the VSD activation and pore opening of the IKs channel is highly recommended to support the idea that CP1 mediates the VSD-pore coupling, just like what the authors did to KCNQ1 alone in Fig 3h.
10. Fig 6. Is 10 μ M CP1 enough or sufficient to see its effects on all the voltage-gated channels here? Dose-response relations should be included, otherwise it seems too strong to state that "CP1 shows no effects on some ion channels other than KCNQ that are voltage dependent" in Line 312. Or modify

statement to "upto tested dose no effect seen".

11. Line 238. Spell out what is the difference between guinea pig and human IKS channel response.
12. Line 298. Rank order does not seem correct. See Q6 above.
13. Fig 5d should be shown with log scale on Y axis. It is impossible to see any effects for small values of tau.

Minor comments:

14. Line 108, maybe label the head of PIP2 in Fig 1a? Also, maybe talk more about similarities and differences between PIP2 and CP1 structures in the manuscript? Please make the PIP2 and CP1 structures at the same scale in Fig 1a for easier comparison between placement of chemical groups.
15. Fig. 1b. This figure is fuzzy and hard to see. Can you make it easier to see the interactions?
16. Fig. 1c. Hard to see the effects due to so many overlapping curves. Can you try some variants to see if it makes it clearer? Maybe thinner lines for ctrl or dashed lines?
17. Fig. 1d. This should really be comparisons of DeltaG, not V0.5.
18. Fig 2c. What cause the funny shape of the CP1 currents?
19. Fig. 2e. Rescue traces do not look like IKS. IS it just constitutive currents?
20. Fig 3h&i. Is the fluorescence really measured correctly? What happen if you try to really move down all S4 before each voltage step, e.g. by a very negative holding potential or prepulse?
21. Fig. 4c&e. It is not clear to this reviewer why you are showing these protocols? Clearly there is some accumulation of open channels. But wouldn't it be more informative to do something to imitate repetitive stimulus as in a heart to see the effects of the drug in a more physiological pulsing?
22. Fig. 4h. Not clear to at all what is shown in this figure. In other figures, Q1 (Fig. 3e) is shifted by 100 mV and Q1/E1 similarly (Fig. 4g). And why start the dose response curve at 7mV?
23. Many figures have strange tick marks at odd intervals (like in Fig 4h). Please make tick marks at more natural places such as every 10 mV instead of every 12 mV (e.g. in 5c, 6b, and 7c).
24. Fig 7g. Please mention the number N for the experiment done in each group.
25. Have the authors applied CP1 and PIP2 together and seen how they affect the KCNQ channels activation? Are they competing with each other or additive?
26. Fig 6. The tail current of CaV1.2 with 10 μ M CP1 is gone, why?
27. Fig 4i. The Y-axis should be negative?
28. Line 308-309. "This study suggests that the interaction of CP1 could similarly distort VSD-pore coupling to such a large extent". I don't understand what this sentence mean. I thought CP1 restores coupling, not distort. And what is large extent?
29. Line 309-310. "is in a delicate balance" You probably mean "has a weak coupling".
30. Line 323. "but any phosphate group that can bind to the channel, such as CP1, will be able to activate the channel" According to Fig 1, CP1 has no phosphate group....
31. Line 327-329. "It is worth noting that all lipid phosphates examined in the previous study did not appear to alter channel activation except for changing the maximal channel activity, while CP1 interaction changes VSD activation and VSD-pore coupling". Did these other studies do VCF, or how did they study VSD activation or VSD-pore coupling?
32. Line 360. Replace frog with Xenopus.

Reviewer #2 (Remarks to the Author):

In this manuscript, the authors identified a small molecule compound, CP1, by screening in silico compounds resembling PIP2 head group to dock onto the structure model of KCNQ1, by targeting the previously identified PIP2 site in a pocket comprising the S4-S5 linker and the S6 C-terminus of the channel. The docking reveals that CP1 and PIP2 each interact with a distinct set of KCNQ1 residues; however, both molecules share some of these interacting residues, including K354 and K358 in the S6

C-terminus. CP1 modifies the activation of KCNQ1 by shifting the voltage dependence of channel opening to more negative potentials. CP1 rescues KCNQ1 currents after PIP2 depletion induced by the Ci-VSP voltage-dependent phosphatase. Using voltage-clamp fluorometry (VCF), the authors found that CP1 enhances VSD-pore coupling and VSD activation of KCNQ1. However, the results suggest that while CP1 acts similarly to PIP2 in that it mediates VSD-pore coupling in KCNQ1 channels, its function may differ from that of PIP2, which does not affect VSD activation or open the pore without VSD activation. It was found that in comparison to KCNQ1 expressed alone, CP1 caused a larger shift of the G-V relationships when KCNE1 was coexpressed. Similar to the effects on KCNQ1 and IKs channels, CP1 increased the amplitude of the currents, shifted the G-V relation to more negative voltages and slowed the deactivation kinetics of KCNQ2 and KCNQ3 channels. In guinea-pig ventricular cardiomyocytes, CP1 reduces drug-induced action potential prolongation.

This study is very interesting and has significant relevance because the results provide insights on PIP2 and CP1 activation of KCNQ channels and a basis for future development of antiarrhythmic drugs that target cardiac KCNQ1 (IKs) channels. The experiments are carefully performed and the results are clearly presented. Nonetheless, I have few concerns, which need to be addressed by the authors: 1-It is not always clear whether CP1 was perfused externally or injected into the oocytes. For example, in figure 1, this information is not provided. In addition, why in some cases it is injected and in others it is applied externally?

2-In Figure 1, the mutations of the KCNQ1 residues and notably the double mutants that interact with CP1 in docking simulations only partially reduced the shift of the G-V relationship. Does it mean that CP1 acts as an "opener" to other interaction sites?

3-In contrast to PIP2, CP1 affects VSD activation and opens the pore without VSD activation. Can the authors discuss somewhat this issue? Is it possible that CP1 acts in a similar way as Polyunsaturated Fatty Acid Analogs (PUFAs) on VSD and S6 residues of KCNQ1? (see Liin et al, 2018. Cell Reports 24, 2908-2918).

4- In Figure 2a, it would be nice to have with the same protocol a control trace without Ci-VSP expression in the absence and presence of CP1.

Reviewer #3 (Remarks to the Author):

The authors describe development of CP1, a PIP2 derivative, and show that it shifts the voltage dependence of KCNQ1, KCNQ1/KCNE1 and KCNQ3 channels to more negative membrane potentials. they also demonstrate that CP1 can be used to correct drug-induced action potential prolongation in guinea-pig ventricular myocytes, which express both IKs and IKr as do human ventricular myocytes. The authors also provide evidence of the location of CP1 binding; indeed, KCNQ1 and PIP2 structures were used to predict CP1 as a possible channel regulator. The work is clearly described and the experiments appear well conducted. CP1 is not highly potent but it is efficacious, producing >80 mV negative shifts in V1/2 of activation at 50 uM. The work will be of interest to others working on KCNQ channel pharmacology and on possible LQT pharmacotherapy development.

I have several specific points:

1) Discussion line 257: what is "PGD"?

2) Generally in the figures: it would be useful to have graphics showing the voltage protocols, at least at the first example of use.

3) Can the authors speculate on what about HCN4 PIP2 sensitivity makes it also react to CP1 like the sensitive KCNQs, while other PIP2-sensitive channels (in and out of the KCNQ subfamily) do not?

4) Figure 2e (upper and lower) - I suggest coloring each individual trace differently and with a key so that the rundown and runup are easier to track. This looks too much like a regular voltage family.

5) Figures 3a, f; 4e; 5a; 6a; 7a - the lines are so thick that the tails cannot be resolved - either add a close-up view or thin the lines.

6) Figure 7a: I suggest referring to the lower traces as "C293B-sensitive control" and "C293B-sensitive CP1" currents because "+C293B" is actually the opposite of what they are (if I am understanding correctly).

7) Figure 7f: it would be nice to see action potentials with CP1 + C293B to show no correction (as a control to show that the CP1 is working via IKs and not other channels).

Referee expertise:

Referee #1: potassium channels, pharmacology

Referee #2: structure/function, potassium channels

Referee #3: K channels, molecular pharmacology

Reviewers' comments:

Reviewer #1 (Remarks to the Author):

The authors here identify CP1 as a substitute for PIP2 mediating the VSD-pore coupling in KCNQ1 channels. CP1 rescued KCNQ1 currents when membrane PIP2 was depleted. Similar to PIP2, CP1 interacts with several amino acids that have been shown to be very important for VSD-pore coupling. However, compare to PIP2 that does not affect the VSD activation or pore opening in the absence of VSD activation, CP1 enhanced the VSD-pore coupling and pore opening in the absence of VSD activation of KCNQ1 channel. Also, the authors found that CP1 had a similar effect on mediating VSD-pore coupling in all KCNQ channels including changing the current amplitude and shifting the G-V relation to more negative voltages. But CP1 showed no effects on other PIP2-sensitive ion channels such as Kir1.1, Cav1.2 and HERG channels. In addition, CP1 shortened the action potential prolongation caused by drug-induced prolongation of ventricular myocytes. By testing a compound CP1 that can mediate the voltage dependent activation of KCNQ1 channels and modulate the drug-induced action potential in ventricular myocytes, the authors conclude that this work helps to develop a potential approach for anti-arrhythmic treatment. The paper is well-organized and easy to read. The experimental data seem convincing and the conclusion seems reasonable. It would attract readers who are interested in the VSD-pore coupling of voltage-gated ion channels and in developing new compounds for the therapy of channelopathies. However, I would like some more discussion with alternative explanations, as listed below.

We thank the reviewer for positive assessment of the work and the thorough review that helps improve the paper.

Major comments:

1. Line 167-169. I don't think these papers are saying that a large fraction of channels be open when only small fraction of channels have moved their S4? How do you explain a FV at more positive voltages than the GV? At least provide a model simulation that gives this phenomena.

This is a good point. The papers cited here suggest that a larger shift of the G-V than Q-V relation is an indication of change in the VSD-pore coupling. As suggested by the reviewer, we now provide a model simulation (supplementary Fig. 3) to illustrate the concept that a change in the VSD-pore coupling can result in a larger shift of G-V than F-V. In the revised manuscript we break the sentence into two to separate what was suggested by previous papers and the model simulation, and what is suggested by the data presented in this manuscript. "A larger shift in G-V

than F-V indicates that CP1 enhanced VSD-pore coupling^{45,46} (Supplementary Fig. 3). Our results show that a small fraction of VSD activation at negative voltages induces a large fraction of pore opening.”

Our model simulation in Supplementary Fig. 3 also shows that the G-V can be at more negative voltages than F-V, which is consistent with the experimental observation. This is a simplified model that does not consider that the channel is a tetramer, containing 4 VSDs, and is only used for proof of concepts. This model and our experimental data are not sufficient for us to fit the model to data and come up with a detailed kinetic description of CP1 action. In previous studies of Ca^{2+} and voltage gated BK channel activation, high Ca^{2+} concentrations were shown to shift G-V to more negative voltages than Q-V (Stefani et al., *PNAS* 94:5427-31, 1997; Horrigan and Aldrich *JGP* 120:267-305, 2002). The mechanism for Ca^{2+} and voltage dependent activation is described by a comprehensive allosteric model (Horrigan and Aldrich *JGP* 120:267-305, 2002).

2. Fig 2f. Why the run down kinetics seems much faster than it in a previous study (Li *et al*, 2011)?

The rundown kinetics depends on internal ATP level, and ATP slows down rundown kinetics (Li et al, *PNAS* 2013). In this study, we did not add ATP to the internal solution, while in the previous study (Li et.al, *PNAS* 2011), we added 1.5 mM MgATP to the internal solution, which helped slow down the rundown kinetics of I_{Ks} .

3. Line 171, is the constitutively conducting current K current? Could it be endogenous current or other currents?

We used control oocytes without injecting channel RNA (uninjected) in our experiments to ensure that the recorded currents were derived from the exogenous KCNQ1 channel expression. We show here the currents in response to one of the voltage pulses (to -120 mV) in the voltage protocol (see Fig. 3a in the manuscript) that measured G-V relation of the KCNQ1 channel in control and CP1 solutions (Revised Fig. 1, rFig. 1). In the presence of CP1 the expressed KCNQ1 currents showed an increased instantaneous current upon the voltage returned from the testing potential of -120 mV to -40 mV. We used the tail current amplitude at -40 mV to construct G-V relations, which showed the constitutively conduction (Fig. 3c in the manuscript).

rFig. 1. Comparison of currents recorded from KCNQ1-expressing (red) and uninjected oocytes (black) at -120 mV (inset: voltage protocol).

4. Line 171-174. I think this study shows the opposite, increasing the Gmin of the pore shifts the GV by allowing relatively more S4 movement in open channels instead of in closed channels. By microreversibility, the S4 movement happens in open channels at more hyperpolarized voltages than in closed channels. I have to point out that the equation given in the paper by Ma

et al. is incorrect and due to faulty mathematical rearrangements of the equations. There is no dependence of G_{min} on $V_{0.5}$: $G_{min} = 1/(1+K)$, where K is the equilibrium constant between C and O with S_4 down.

The reviewer makes a good point. We revised the manuscript to not connect the constitutive opening at negative voltages with VSD-pore coupling.

5. Line 179. Could CP1 have effects on the pore independent of coupling? This should at least be entertained.

This is a good suggestion and we now separate the constitutive opening at negative voltages with VSD-pore coupling (see our response to the previous comment.)

6. Line 187. "coexpression of KCNQ1 with KCNE1 exhibited a stronger response to CP1 than KCNQ1 alone." Is this really true? Seems like voltage shift is bigger in KCNQ1 alone? How do you quantify stronger response? Maybe this is just due to your inability to hold negative enough to close completely channels between pulses? Some quantitative effect on a channel parameter would be best to show, i.e. Delta $V_{0.5}$, G_{max} increase, etc.

This statement is a summary of the effects of CP1 on KCNQ1+KCNE1 described afterwards. These effects appear to be stronger than the effects of CP1 on KCNQ1 alone. These effects include a slower deactivation, a larger shift of G-V, and larger current amplitudes. The quantitative effect on channel parameters are shown and compared between KCNQ1 alone and KCNQ1+KCNE1 in Fig. 4, including Delta $V_{0.5}$ (Fig. 4i), G_{max} increase (Fig. 4h), etc. The reviewer may be right that these effects were due to our inability to hold negative enough to close completely channels between pulses. However, we were able to close completely KCNQ1 channels alone. This fact suggests that KCNQ1 co-expression with KCNE1 shows a stronger response to CP1 than KCNQ1 alone.

7. Line 193, if after treating with 2 μ M CP1, the IKs channels show a big fraction of constitutive currents at the very negative voltages, why it is not reflected in the GV relation in Fig 4d.

Thanks for pointing this out. The G-V relation in the original Fig. 4d was scaled to 0 at negative voltages in order to illustrate the G-V shift. In the revised manuscript the unscaled G-V is used as Fig. 4d, which show constitutive activation.

8. Line 204. "suggesting additional IKs channels that had been silent in the absence of CP1 were activated by CP1.". Maybe just a change in maximum open probability?

This is an excellent point. The sentence is changed to include this possibility.

9. A VCF experiment showing the effects of CP1 on the VSD activation and pore opening of the IKs channel is highly recommended to support the idea that CP1 mediates the VSD-pore coupling, just like what the authors did to KCNQ1 alone in Fig 3h.

The results of Fig. 2 indicate that CP1 mediates the VSD-pore coupling of the KCNQ1 channel either with or without KCNE1 association. The results in Fig. 3 and 4 show that CP1 alters function of KCNQ1 alone and of KCNQ1 + KCNE1 similarly. However, we agree with the reviewer that a VCF experiment would provide more direct evidence to support that in I_{Ks} channels CP1 binding also enhances VSD-pore coupling when the native PIP_2 in the membrane is not depleted. We had tried to do the experiment before submitting the manuscript and tried again after receiving this comment. It is known that VCF experiments have a lower success rate for the I_{Ks} channel than for the KCNQ1 channel alone. CP1 made the experiment even harder. We did not succeed in getting the data from these experiments.

10. *Fig 6. Is 10 μ M CP1 enough or sufficient to see its effects on all the voltage-gated channels here? Dose-response relations should be included, otherwise it seems too strong to state that “CP1 shows no effects on some ion channels other than KCNQ that are voltage dependent” in Line 312. Or modify statement to “upto tested dose no effect seen”.*

As suggested, we modified the sentence by specifying that “10 μ M CP1 shows no effect...”. Thanks for this suggestion, which makes the statement more precise.

11. *Line 238. Spell out what is the difference between guinea pig and human I_{Ks} channel response.*

The comparison between the KCNQ1 expressed in oocytes and I_{Ks} in guinea pig myocytes is now provided.

12. *Line 298. Rank order does not seem correct. See Q6 above.*

The rank order is consistent with data shown in Fig. 4i and Fig. 5c.

13. *Fig 5d should be shown with log scale on Y axis. It is impossible to see any effects for small values of tau.*

Done as suggested.

Minor comments:

14. *Line 108, maybe label the head of PIP_2 in Fig 1a? Also, maybe talk more about similarities and differences between PIP_2 and CP1 structures in the manuscript? Please make the PIP_2 and CP1 structures at the same scale in Fig 1a for easier comparison between placement of chemical groups.*

Thanks for the valuable suggestion. In Fig. 1a we label the head of PIP_2 , resize and re-orient CP1 for easier structural comparison, and circle the negatively charged residues (see the revised Fig. 1a and figure caption). The similarities and differences between PIP_2 and CP1 on their interacting modes with KCNQ1 are described in the revised manuscript.

15. *Fig. 1b. This figure is fuzzy and hard to see. Can you make it easier to see the interactions?*

Fig. 1b has been re-plotted. A new supplementary figure (Supplementary Fig. 1) has been added to present the details of the interactions of PIP₂ and CP1 with KCNQ1, respectively.

16. *Fig. 1c. Hard to see the effects due to so many overlapping curves. Can you try some variants to see if it makes it clearer? Maybe thinner lines for ctrl or dashed lines?*

We make the symbols smaller and lines thinner as suggested.

17. *Fig. 1d. This should really be comparisons of DeltaG, not V0.5.*

We used G-V shift as a measure of CP1 effect in this experiment, and DeltaV0.5 indicates G-V shift.

18. *Fig 2c. What cause the funny shape of the CP1 currents?*

This is a good question, but we do not know the answer. In most of our recordings, the CP1 currents showed similar shapes so that this is a typical result. We can only speculate that this shape is due to the transition state at the time of CP1 application when CP1 concentration, PIP₂ concentration and the binding-unbinding of CP1 and PIP₂ by the channel proteins are not in a steady state.

19. *Fig. 2e. Rescue traces do not look like I_{Ks}. IS it just constitutive currents?*

CP1 alters I_{Ks} kinetics as well as current amplitudes (see Fig. 4). It is possible that the rescue traces in Fig. 2e show some similar properties.

20. *Fig 3h&i. Is the fluorescence really measured correctly? What happen if you try to really move down all S4 before each voltage step, e.g. by a very negative holding potential or prepulse?*

For KCNQ1 channels the deactivation rate is well within the interval between activation pulses (Fig. 3D) and we did not observe accumulation of activation of the VSD (Fig. 3i). Based on these data we think that a very negative holding potential or prepulse may not cause significant change of the results as presented. VCF experiments are generally more difficult to complete. We determined the experimental protocol based on the consideration of scientific rigor as well as practicality.

21. *Fig. 4c&e. It is not clear to this reviewer why you are showing these protocols? Clearly there is some accumulation of open channels. But wouldn't it be more informative to do something to imitate repetitive stimulus as in a heart to see the effects of the drug in a more physiological pulsing?*

We show these currents to illustrate the degree of activation accumulation under our experimental conditions and the impact of the accumulation on the measurements of activation parameters. The current properties depend on the ratio of KCNQ1:KCNE1, which cannot be accurately simulated by the expression of the channel in oocytes. In this figure we provide biophysical properties of the channels, and in subsequent experiments (Fig. 7) we test CP1 effects on guinea pig myocytes.

22. *Fig. 4h. Not clear to at all what is shown in this figure. In other figures, Q1 (Fig. 3e) is shifted by 100 mV and Q1/E1 similarly (Fig. 4g). And why start the dose response curve at 7mV?*

Thanks for pointing these out. Fig. 4h is to show the relative increase of conductance of the channels at all voltages caused by CP1. To make it more clear, we describe the procedure for making the plot in revised legend. The Fig. 4i Y-axis was labeled wrong by mistake. We have corrected label in the revised figure.

23. *Many figures have strange tick marks at odd intervals (like in Fig 4h). Please make tick marks at more natural places such as every 10 mV instead of every 12 mV (e.g. in 5c, 6b, and 7c).*

Thanks for bringing up this good point. We have changed all the ticks in fig. 4h, 5c (every -50 mV on Y-axis), 6b (every 50 μ A on Y-axis) and 7c (every 5 pA/pF on Y-axis) as suggested.

24. *Fig 7g. Please mention the number N for the experiment done in each group.*

Added.

25. *Have the authors applied CP1 and PIP2 together and seen how they affect the KCNQ channels activation? Are they competing with each other or additive?*

No, we did not, but this would be a good experiment for future studies. When we applied CP1 to oocytes expressing the channels the native PIP₂ in the oocyte membrane was present. CP1 altered channel properties, which seems to suggest that CP1 may compete with PIP₂ or bind to the site that is not occupied by PIP₂ to alter channel function. However, more systematic studies are needed to answer this question.

26. *Fig 6. The tail current of CaV1.2 with 10 μ M CP1 is gone, why?*

The tail currents are prominent in some recordings but not in others, which is similar whether in the presence or absence of CP1. We do not know the reason for these variations or if CP1 has any real effect on the tail currents.

27. *Fig 4i. The Y-axis should be negative?*

The Y-axis in Fig. 4i is changed with negative value.

28. Line 308-309. "This study suggests that the interaction of CP1 could similarly distort VSD-pore coupling to such a large extent". I don't understand what this sentence mean. I thought CP1 restores coupling, not distort. And what is large extent?

These are good points. The text is revised to be more precise.

29. Line 309-310. "is in a delicate balance" You probably mean "has a weak coupling".

The coupling could be weak but we are not sure if it is accurate. The sentence is revised to simply point out the observations without defining the nature of the coupling.

30. Line 323. "but any phosphate group that can bind to the channel, such as CP1, will be able to activate the channel" According to Fig 1, CP1 has no phosphate group....

The reviewer is right, thanks. The sentence is changed.

31. Line 327-329. "It is worth noting that all lipid phosphates examined in the previous study did not appear to alter channel activation except for changing the maximal channel activity, while CP1 interaction changes VSD activation and VSD-pore coupling". Did these other studies do VCF, or how did they study VSD activation or VSD-pore coupling?

No, they did not do VCF, but they recorded ionic currents and found no change in activation properties.

32. Line 360. Replace frog with *Xenopus*.

Done.

Reviewer #2 (Remarks to the Author):

In this manuscript, the authors identified a small molecule compound, CP1, by screening in silico compounds resembling PIP2 head group to dock onto the structure model of KCNQ1, by targeting the previously identified PIP2 site in a pocket comprising the S4-S5 linker and the S6 C-terminus of the channel. The docking reveals that CP1 and PIP2 each interact with a distinct set of KCNQ1 residues; however, both molecules share some of these interacting residues, including K354 and K358 in the S6 C-terminus. CP1 modifies the activation of KCNQ1 by shifting the voltage dependence of channel opening to more negative potentials. CP1 rescues KCNQ1 currents after PIP2 depletion induced by the Ci-VSP voltage-dependent phosphatase. Using voltage-clamp fluorometry (VCF), the authors found that CP1 enhances VSD-pore coupling and VSD activation of KCNQ1. However, the results suggest that while CP1 acts similarly to PIP2 in that it mediates VSD-pore coupling in KCNQ1 channels, its function may differ from that of PIP2, which does not affect VSD activation or open the pore without VSD activation. It was found that in comparison to KCNQ1 expressed alone, CP1 caused a larger

shift of the G-V relationships when KCNE1 was coexpressed. Similar to the effects on KCNQ1 and IKs channels, CP1 increased the amplitude of the currents, shifted the G-V relation to more negative voltages and slowed the deactivation kinetics of KCNQ2 and KCNQ3 channels. In guinea-pig ventricular cardiomyocytes, CP1 reduces drug-induced action potential prolongation. This study is very interesting and has significant relevance because the results provide insights on PIP2 and CP1 activation of KCNQ channels and a basis for future development of antiarrhythmic drugs that target cardiac KCNQ1 (IKs) channels. The experiments are carefully performed and the results are clearly presented. Nonetheless, I have few concerns, which need to be addressed by the authors:

We appreciate the positive comments of the reviewer and the following comments that help improve the manuscript.

1-It is not always clear whether CP1 was perfused externally or injected into the oocytes. For example, in figure 1, this information is not provided. In addition, why in some cases it is injected and in others it is applied externally?

In Fig. 2a, b CP1 was injected into the oocytes. While in all the other figures, CP1 was applied in bath solution. We now add a statement of this fact at the end of Fig. 1 legend.

The experiments in Fig. 2a, b were performed at the early stage of the project, when we did not realize that CP1 was permeable to the membrane.

2-In Figure 1, the mutations of the KCNQ1 residues and notably the double mutants that interact with CP1 in docking simulations only partially reduced the shift of the G-V relationship. Does it mean that CP1 acts as an “opener” to other interaction sites?

This is a good question. From current results, the double MUTATION R249A/S253A, R249A/K354A and K354A/K358A reduced the GV shift caused by CP1. And the triplicate mutation R249A/S253A/K354A/K358A further reduced the GV shift. The interactions between the compound and channel protein may not be abolished completely by site directed mutations due to the flexibility of the compound and the involvement of multiple chemical groups. On the other hand, these results cannot exclude the possibility that CP1 may act as an opener by binding to some other sites as the reviewer suggested. We acknowledged this possibility by concluding that the identified site interacts with CP1, and in Discussion we stated that we do not know if CP1 shows any state-dependent binding to different sites. We now discuss further this possibility (see response to question 3 of the same reviewer).

3-In contrast to PIP2, CP1 affects VSD activation and opens the pore without VSD activation. Can the authors discuss somewhat this issue? Is it possible that CP1 acts in a similar way as Polyunsaturated Fatty Acid Analogs (PUFAs) on VSD and S6 residues of KCNQ1? (see Liin et al, 2018. Cell Reports 24, 2908-2918).

Thanks for bringing up this possibility. We discuss this possibility as suggested at the end of the second paragraph of Discussion.

4- In Figure 2a, it would be nice to have with the same protocol a control trace without Ci-VSP expression in the absence and presence of CP1.

The requested data are shown below (Revised Fig. 2, rFig. 2). The results are similar as in Figure

rFig 2. KCNQ1 currents were recorded at +40 mV (the protocol at top). After the first trace (black), the KCNQ1 current amplitude fluctuated. The same voltage pulses were applied continuously until the current became stable (gray, ~ 10 pulses from the first trace). Then we applied 10 μ M CP1 and the currents would decrease to a stable state (red).

3a and, therefore, are not shown in Fig. 2.

Reviewer #3 (Remarks to the Author):

The authors describe development of CP1, a PIP2 derivative, and show that it shifts the voltage dependence of KCNQ1, KCNQ1/KCNE1 and KCNQ3 channels to more negative membrane potentials. they also demonstrate that CP1 can be used to correct drug-induced action potential prolongation in guinea-pig ventricular myocytes, which express both IKs and IKr as do human ventricular myocytes. The authors also provide evidence of the location of CP1 binding; indeed, KCNQ1 and PIP2 structures were used to predict CP1 as a possible channel regulator. The work is clearly described and the experiments appear well conducted. CP1 is not highly potent but it is efficacious, producing >80 mV negative shifts in V_{1/2} of activation at 50 μ M. The work will be of interest to others working on KCNQ channel pharmacology and on possible LQT pharmacotherapy development.

We would like to thank the reviewer for recognizing the significance of the results and the comments to help improving the manuscript.

I have several specific points:

1) *Discussion line 257: what is "PGD"?*

PGD is the abbreviation of "Pore-gated Domain". It is changed to "pore".

2) *Generally in the figures: it would be useful to have graphics showing the voltage protocols, at least at the first example of use.*

This is a good suggestion. We add voltage protocols in Fig. 2 and Fig. 3a.

3) *Can the authors speculate on what about HCN4 PIP2 sensitivity makes it also react to CP1*

like the sensitive KCNQs, while other PIP₂-sensitive channels (in and out of the KCNQ subfamily) do not?

Positively charged residues are important in PIP₂ binding sites in proteins. These residues do not have a consensus sequence and the sites differ in different proteins. It is possible that the positively charged residues in the KCNQ1 and HCN4 channels are arranged in a way that they also interact with CP1 but not in other channels. It is also possible that CP1 may happen to fit the PIP₂ sites pocket in KCNQ1 and HCN4 channels better than in other channels. However, we do not have any evidence to answer this question. It may be answered by future studies that provide experimental evidence.

4) Figure 2e (upper and lower) - I suggest coloring each individual trace differently and with a key so that the rundown and runup are easier to track. This looks too much like a regular voltage family.

Done as suggested.

5) Figures 3a, f; 4e; 5a; 6a; 7a - the lines are so thick that the tails cannot be resolved - either add a close-up view or thin the lines.

Thanks for the suggestion. We make the lines thinner and the figures are resolved better.

6) Figure 7a: I suggest referring to the lower traces as "C293B-sensitive control" and "C293B-sensitive CP1" currents because "+C293B" is actually the opposite of what they are (if I am understanding correctly).

Good suggestion, done.

7) Figure 7f: it would be nice to see action potentials with CP1 + C293B to show no correction (as a control to show that the CP1 is working via IKs and not other channels).

We have performed additional experiments as suggested, and the results are inserted as Fig. 7f in revised Fig. 7.

REVIEWERS' COMMENTS:

Reviewer #1 (Remarks to the Author):

The authors have responded well to my comments. I only have one minor issue.

Lines 185-187: "Similar to our observation, a previous study showed a constitutive opening of KCNQ1, which increased when the G-V relation was shifted to more negative voltages by mutations⁴⁸." I think this sentence is somewhat misleading. In my mind, it is really the other way around. These mutations in Ma et al were in the pore and most likely these mutations shifted the stability of the open/closed conformation, which indirectly shifted the GV. So it is a question of which came first: the chicken (pore effect) or the egg (GV shift). In this case, I would vote for the chicken (pore).

Reviewer #2 (Remarks to the Author):

The revised manuscript has been improved and addressed the concerns of this reviewer.

Reviewer #3 (Remarks to the Author):

The authors have addressed my previous concerns with edits to the figures and text, and additional experiments.